# Censored Sampling of Diffusion Models Using 3 Minutes of Human Feedback

**TaeHo Yoon**[1]  **Kibeom Myoung**[1]  **Keon Lee**[4]  **Jaewoong Cho**[4]  **Albert No**[3]  **Ernest K. Ryu**[1,2]

[1]Department of Mathematical Science, Seoul National University
[2]Interdisciplinary Program in Artificial Intelligence, Seoul National University
[3]Department of Electronic and Electrical Engineering, Hongik University
[4]KRAFTON

## Abstract

Diffusion models have recently shown remarkable success in high-quality image generation. Sometimes, however, a pre-trained diffusion model exhibits partial misalignment in the sense that the model can generate good images, but it sometimes outputs undesirable images. If so, we simply need to prevent the generation of the bad images, and we call this task censoring. In this work, we present censored generation with a pre-trained diffusion model using a reward model trained on minimal human feedback. We show that censoring can be accomplished with extreme human feedback efficiency and that labels generated with a mere few minutes of human feedback are sufficient. Code available at: `https://github.com/tetrzim/diffusion-human-feedback`.

## 1 Introduction

Diffusion probabilistic models [19, 12, 42] have recently shown remarkable success in high-quality image generation. Much of the progress is driven by scale [35, 36, 38], and this progression points to a future of spending high costs to train a small number of large-scale foundation models [4] and deploying them, sometimes with fine-tuning, in various applications. In particular use cases, however, such pre-trained diffusion models may be misaligned with goals specified before or after the training process. An example of the former is text-guided diffusion models occasionally generating content with nudity despite the text prompt containing no such request. An example scenario of the latter is deciding that generated images should not contain a certain type of concepts (for example, human faces) even though the model was pre-trained on images with such concepts.

Fixing misalignment directly through training may require an impractical cost of compute and data. To train a large diffusion model again from scratch requires compute costs of up to hundreds of thousands of USD [30, 29]. To fine-tune a large diffusion model requires data size ranging from 1,000 [28] to 27,000 [25].[1] We argue that such costly measures are unnecessary when the pre-trained model is already capable of sometimes generating "good" images. If so, we simply need to prevent the generation of "bad" images, and we call this task *censoring*. (Notably, censoring does not aim to improve the "good" images.) Motivated by the success of reinforcement learning with human feedback (RLHF) in language domains [9, 50, 43, 33], we perform censoring using human feedback.

In this work, we present censored generation with a pre-trained diffusion model using a reward model trained on extremely limited human feedback. Instead of fine-tuning the pre-trained diffusion model, we train a reward model on labels generated with a **few minutes of human feedback** and perform guided generation. By not fine-tuning the diffusion model (score network), we reduce both compute

---

[1]The prior work [28] fine-tunes a pre-trained diffusion model on a new dataset of size 1k using a so-called adapter module while [25] improves text-to-image alignment using 27k human-feedback data.

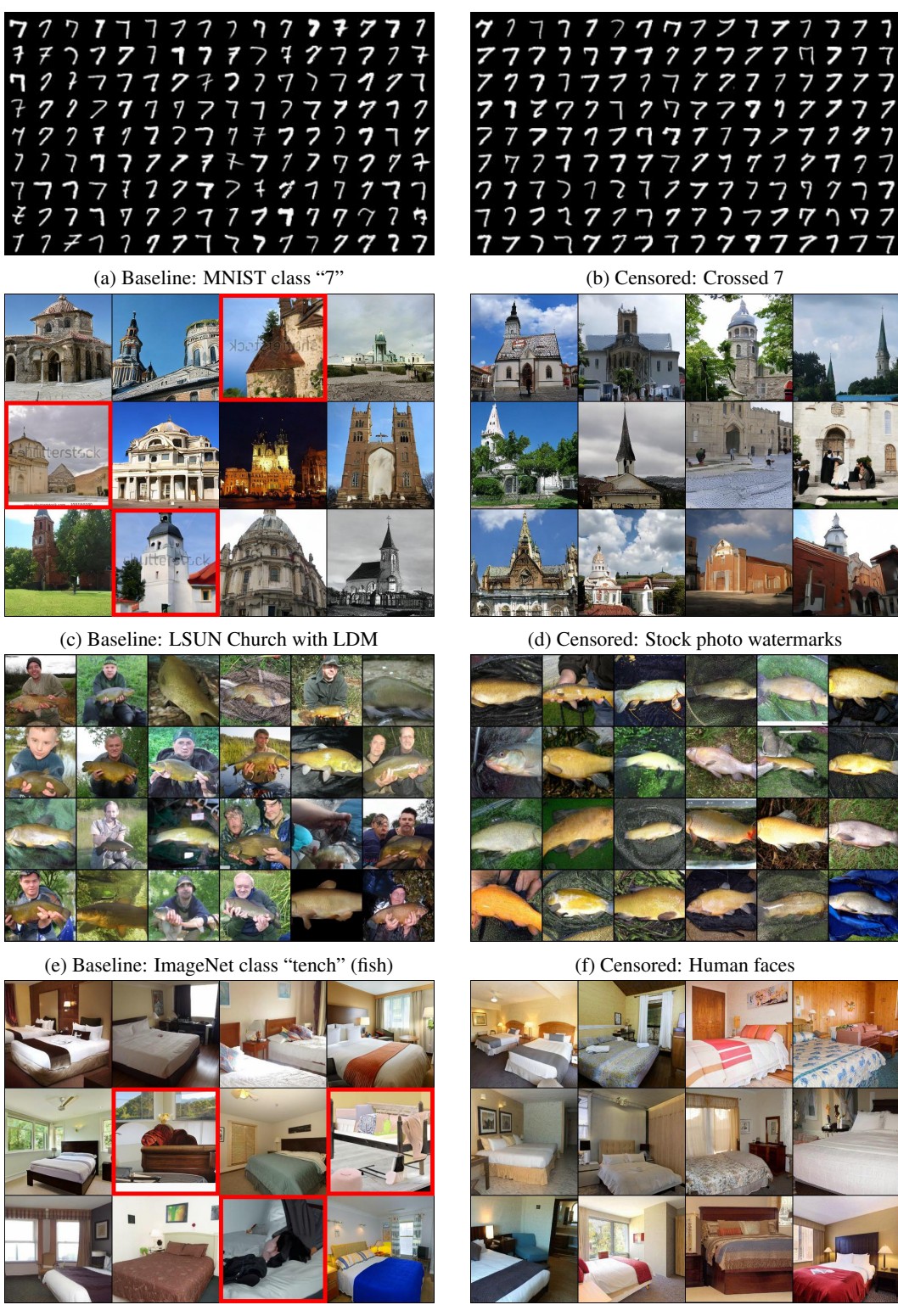

(a) Baseline: MNIST class "7"

(b) Censored: Crossed 7

(c) Baseline: LSUN Church with LDM

(d) Censored: Stock photo watermarks

(e) Baseline: ImageNet class "tench" (fish)

(f) Censored: Human faces

(g) Baseline: LSUN bedroom

(h) Censored: Broken images

Figure 1: Uncensored baseline vs. censored generation. Setups are precisely defined in Section 5. Due to space constraints, we present selected representative images here. Full sets of non-selected samples are shown in the appendix.

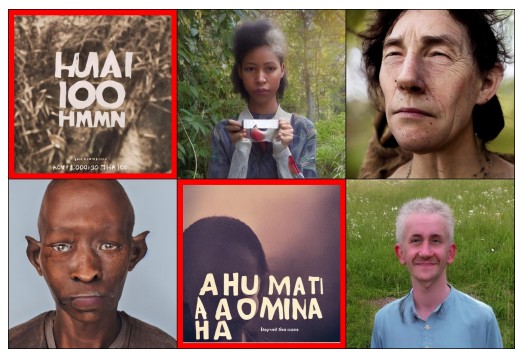

(i) Baseline: Stable Diffusion, "A photo of a human"

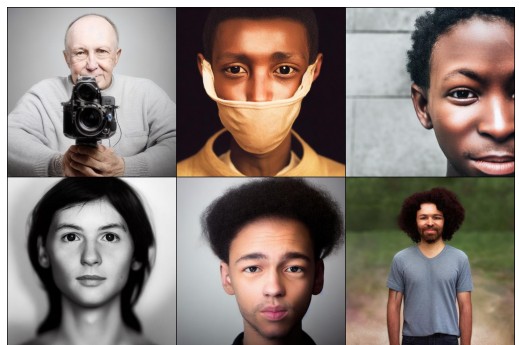

(j) Censored: Embedded text

Figure 1: (Continued) Uncensored baseline vs. censored generation. Setups are precisely defined in Section 5. Due to space constraints, we present selected representative images here. Full sets of non-selected samples are shown in the appendix.

and data requirements for censored generation to negligible levels. (Negligible compared to any amount of compute and man-hours an ML scientist would realistically spend building a system with a diffusion model.) We conduct experiments within multiple setups demonstrating how minimal human feedback enables removal of target concepts. The specific censoring targets we consider are: A handwriting variation ("crossed 7"s) in MNIST [11]; Watermarks in the LSUN [47] church images; Human faces in the ImageNet [10] class "tench"; "Broken" images in the generation of LSUN bedroom images.

**Contribution.** Most prior work focus on training new capabilities into diffusion models, and this inevitably requires large compute and data. Our main contribution is showing that a very small amount of human feedback data and computation is sufficient for guiding a pre-trained diffusion model to do what it can already do while suppressing undesirable behaviors.

## 1.1 Background on diffusion probabilistic models

Due to space constraints, we defer the comprehensive review of prior works to Appendix D. In this section, we briefly review the standard methods of diffusion probabilistic models (DPM) and set up the notation. For the sake of simplicity and specificity, we only consider the DPMs with the variance preserving SDE.

Consider the *variance preserving (VP)* SDE

$$dX_t = -\frac{\beta_t}{2}X_t dt + \sqrt{\beta_t}dW_t, \qquad X_0 \sim p_0 \tag{1}$$

for $t \in [0, T]$, where $\beta_t > 0$, $X_t \in \mathbb{R}^d$, and $W_t$ is a $d$-dimensional Brownian motion. The process $\{X_t\}_{t \in [0,T]}$ has the marginal distributions given by

$$X_t \overset{\mathcal{D}}{=} \sqrt{\alpha_t}X_0 + \sqrt{1-\alpha_t}\varepsilon_t, \qquad \alpha_t = e^{-\int_0^t \beta_s ds}, \ \varepsilon_t \sim \mathcal{N}(0, I)$$

for $t \in [0, T]$ [39, Chapter 5.5]. Let $p_t$ denote the density for $X_t$ for $t \in [0, T]$. Anderson's theorem [1] tells us that the the reverse-time SDE by

$$d\overline{X}_t = \beta_t \left( -\nabla \log p_t(\overline{X}_t) - \frac{1}{2}\overline{X}_t \right) dt + \sqrt{\beta_t}d\overline{W}_t, \qquad \overline{X}_T \sim p_T,$$

where $\{\overline{W}_t\}_{t \in [0,T]}$ is a reverse-time Brownian motion, satisfies $\overline{X}_t \overset{\mathcal{D}}{=} X_t \sim p_t$.

In DPMs, the initial distribution is set as the data distribution, i.e., $p_0 = p_{\text{data}}$ in (1), and a *score network* $s_\theta$ is trained so that $s_\theta(X_t, t) \approx \nabla \log p_t(X_t)$. For notational convenience, one often uses the *error network* $\varepsilon_\theta(X_t, t) = -\sqrt{1-\alpha_t}s_\theta(X_t, t)$. Then, the reverse-time SDE is approximated by

$$d\overline{X}_t = \beta_t \left( \frac{1}{\sqrt{1-\alpha_t}}\varepsilon_\theta(\overline{X}_t, t) - \frac{1}{2}\overline{X}_t \right) dt + \sqrt{\beta_t}d\overline{W}_t, \qquad \overline{X}_T \sim \mathcal{N}(0, I)$$

for $t \in [0, T]$.

When an image $X$ has a corresponding label $Y$, classifier guidance [40, 12] generates images from

$$p_t(X_t \mid Y) \propto p_t(X_t, Y) = p_t(X_t)p_t(Y \mid X_t)$$

for $t \in [0, T]$ using

$$\hat{\varepsilon}_\theta(\overline{X}_t, t) = \varepsilon_\theta(\overline{X}_t, t) - \omega\sqrt{1 - \alpha_t}\nabla \log p_t(Y \mid \overline{X}_t)$$

$$d\overline{X}_t = \beta_t \left( \frac{1}{\sqrt{1 - \alpha_t}} \hat{\varepsilon}_\theta(\overline{X}_t, t) - \frac{1}{2}\overline{X}_t \right) dt + \sqrt{\beta_t}d\overline{W}_t, \qquad \overline{X}_T \sim \mathcal{N}(0, I),$$

where $\omega > 0$. This requires training a separate time-dependent classifier approximating $p_t(Y \mid X_t)$. In the context of censoring, we use a reward model with binary labels in place of a classifier.

## 2 Problem description: Censored sampling with human feedback

Informally, our goal is:

> Given a pre-trained diffusion model that is partially misaligned in the sense that generates both "good" and "bad" images, fix/modify the generation process so that only good images are produced.

The meaning of "good" and "bad" depends on the context and will be specified through human feedback. For the sake of precision, we define the terms "benign" and "malign" to refer to the good and bad images: A generated image is *malign* if it contains unwanted features to be censored and is *benign* if it is not malign.

Our assumptions are: (i) the pre-trained diffusion model does not know which images are benign or malign, (ii) a human is willing to provide minimal ($\sim$ 3 minutes) feedback to distinguish benign and malign images, and (iii) the compute budget is limited.

**Mathematical formalism.** Suppose a pre-trained diffusion model generates images from distribution $p_{\text{data}}(x)$ containing both benign and malign images. Assume there is a function $r(x) \in (0, 1)$ representing the likelihood of $x$ being benign, i.e., $r(x) \approx 1$ means image $x$ is benign and should be considered for sampling while $r(x) \approx 0$ means image $x$ is malign and should not be sampled. We mathematically formalize our goal as: Sample from the censored distribution

$$p_{\text{censor}}(x) \propto p_{\text{data}}(x)r(x).$$

**Human feedback.** The definition of benign and malign images are specified through human feedback. Specifically, we ask a human annotator to provide binary feedback $Y \in \{0, 1\}$ for each image $X$ through a simple graphical user interface shown in Appendix E. The feedback takes 1–3 human-minutes for the relatively easier censoring tasks and at most 10–20 human-minutes for the most complex task that we consider. Using the feedback data, we train a *reward model* $r_\psi \approx r$, which we further detail in Section 3.

**Evaluation.** The evaluation criterion of our methodology are the human time spent providing feedback, quantified by direct measurement, and sample quality, quantified by precision and recall.

In this context, *precision* is the proportion of benign images, and *recall* is the sample diversity of the censored generation. Precision can be directly measured by asking human annotators to label the final generated images, but recall is more difficult to measure. Therefore, we primarily focus on precision for quantitative evaluation. We evaluate recall qualitatively by providing the generated images for visual inspection.

## 3 Reward model and human feedback

Let $Y$ be a random variable such that $Y = 1$ if $X$ is benign and $Y = 0$ if $X$ is malign. Define the time-independent reward function as

$$r(X) = \mathbb{P}(Y = 1 \mid X).$$

---

**Algorithm 1** Reward model ensemble

---

**Require:** Images: malign $\{X^{(1)}, \ldots, X^{(N_M)}\}$, benign $\{X^{(N_M+1)}, \ldots, X^{(N_M+N_B)}\}$ $(N_M < N_B)$
    **for** $k = 1, \ldots, K$ **do**
        Randomly select with replacement $N_M$ benign samples $X^{(N_M+i_1)}, \ldots, X^{(N_M+i_{N_M})}$.
        Train reward model $r_{\psi_k}^{(k)}$ with $\{X^{(1)}, \ldots, X^{(N_M)}\} \cup \{X^{(N_M+i_1)}, \ldots, X^{(N_M+i_{N_M})}\}$ .
    **end for**
    **return** $r_\psi = \prod_{k=1}^{K} r_{\psi_k}^{(k)}$

---

As we later discuss in Section 4, time-dependent guidance requires a time-dependent reward function. Specifically, let $X \sim p_{\text{data}}$ and $Y$ be its label. Let $\{X_t\}_{t \in [0,T]}$ be images corrupted by the VP SDE (1) with $X_0 = X$. Define the time-dependent reward function as

$$r_t(X_t) = \mathbb{P}(Y = 1 \mid X_t) \qquad \text{for } t \in [0, T].$$

We approximate the reward function $r$ with a *reward model* $r_\psi$, i.e., we train

$$r_\psi(X) \approx r(X) \qquad \text{or} \qquad r_\psi(X_t, t) \approx r_t(X_t),$$

using human feedback data $(X^{(1)}, Y^{(1)}), \ldots, (X^{(N)}, Y^{(N)})$. (So the time-dependent reward model uses $(X_t^{(n)}, Y^{(n)})$ as training data.) We use weighted binary cross entropy loss. In this section, we describe the most essential components of the reward model while deferring details to Appendix F.

The main technical challenge is achieving extreme human-feedback efficiency. Specifically, we have $N < 100$ in most setups we consider. Finally, we clarify that the diffusion model (score network) is not trained or fine-tuned. We use relatively large pre-trained diffusion models [12, 36], but we only train the relatively lightweight reward model $r_\psi$.

### 3.1 Reward model ensemble for benign-dominant setups

In some setups, benign images constitute the majority of uncensored generation. Section 5.2 considers such a *benign-dominant* setup, where 11.4% of images have stock photo watermarks and the goal is to censor the watermarks. A random sample of images provided to a human annotator will contain far more benign than malign images.

To efficiently utilize the imbalanced data in a sample-efficient way, we propose an ensemble method loosely inspired by ensemble-based sample efficient RL methods [23, 6]. The method trains $K$ reward models $r_{\psi_1}^{(1)}, \ldots, r_{\psi_K}^{(K)}$, each using a shared set of $N_M$ (scarce) malign images joined with $N_M$ benign images randomly subsampled bootstrap-style from the provided pool of $N_B$ (abundant) benign data as in Algorithm 1. The final reward model is formed as $r_\psi = \prod_{k=1}^{K} r_{\psi_k}^{(k)}$. Given that a product becomes small when any of its factor is small, $r_\psi$ is effectively asking for unanimous approval across $r_{\psi_1}^{(1)}, \ldots, r_{\psi_K}^{(K)}$.

In experiments, we use $K = 5$. We use the same neural network architecture for $r_{\psi_1}^{(1)} \ldots, r_{\psi_K}^{(K)}$, whose parameters $\psi_1, \ldots, \psi_K$ are either independently randomly initialized or transferred from the same pre-trained weights as discussed in Section 3.3. We observe that the ensemble method significantly improves the precision of the model without perceivably sacrificing recall.

### 3.2 Imitation learning for malign-dominant setups

In some setups, malign images constitute the majority of uncensored generation. Section 5.3 considers such a *malign-dominant* setup, where 69% of images are tench (fish) images with human faces and the goal is to censor the images with human faces. Since the ratio of malign images starts out high, a single round of human feedback and censoring may not sufficiently reduce the malign ratio.

Therefore, we propose an imitation learning method loosely inspired by imitation learning RL methods such as DAgger [37]. The method collects human feedback data in multiple rounds and improves the reward model over the rounds as described in Algorithm 2. Our experiment of Section 5.3 indicates that 2–3 rounds of imitation learning dramatically reduce the ratio of malign images. Furthermore,

---

**Algorithm 2** Imitation learning of reward model

---

**Require:** Pre-trained $\varepsilon_\theta$. Initialize $\mathcal{D} = \emptyset$.
  Sample $X^{(1)}, \ldots, X^{(N_1)}$ using $\varepsilon_\theta$ and no censoring.
  Receive $Y^{(1)}, \ldots, Y^{(N_1)}$ from human feedback. Add data to buffer: $\mathcal{D} \leftarrow \{(X^{(i)}, Y^{(i)})\}_{i=1}^{N_1}$.
  Train reward model $r_\psi$ with $\mathcal{D}$.
  **for** $r = 2, \ldots, R$ **do**
      Sample $X^{(1)}, \ldots, X^{(N_r)}$ using $\varepsilon_\theta$ and censoring with $r_\psi$.
      Receive $Y^{(1)}, \ldots, Y^{(N_r)}$ from human feedback. Add data to buffer: $\mathcal{D} \leftarrow \{(X^{(i)}, Y^{(i)})\}_{i=1}^{N_r}$.
      Train reward model $r_\psi$ with $\mathcal{D}$.
  **end for**
  **return** $r_\psi$

---

imitation learning is a practical model of an online scenario where one continuously trains and updates the reward model $r_\psi$ while the diffusion model is continually deployed.

**Ensemble vs. imitation learning.**   In the benign-dominant setup, imitation learning is too costly in terms of human feedback since acquiring sufficiently many ($\sim 10$) malign labels may require the human annotator to go through too many benign labels ($\sim 1000$) for the second round of human feedback and censoring. In the malign-dominant setup, one can use a reward model ensemble, where reward models share the benign data while bootstrap-subsampling the malign data, but we empirically observe this to be ineffective. We attribute this asymmetry to the greater importance of malign data over benign data; the training objective is designed so as our primary goal is to censor malign images.

### 3.3   Transfer learning for time-independent reward

To further improve human-feedback efficiency, we use transfer learning. Specifically, we take a ResNet18 model [17, 18] pre-trained on ImageNet1k [10] and replace the final layer with randomly initialized fully connected layers which have 1-dimensional output features. We observe training all layers to be more effective than training only the final layers. We use transfer learning only for training time-independent reward models, as pre-trained time-dependent classifiers are less common. Transfer learning turns out to be essential for complex censoring tasks (Sections 5.2, 5.4 and 5.5), but requires guidance techniques other than the simple classifier guidance (see Section 4).

## 4   Sampling

In this section, we describe how to perform censored sampling with a trained reward model $r_\psi$. We follow the notation of Section 1.1.

**Time-dependent guidance.**   Given a time-dependent reward model $r_\psi(X_t, t)$, our censored generation follows the SDE

$$\hat{\varepsilon}_\theta(\overline{X}_t, t) = \varepsilon_\theta(\overline{X}_t, t) - \omega\sqrt{1 - \alpha_t}\nabla \log r_t(\overline{X}_t)$$

$$d\overline{X}_t = \beta_t\left(\frac{1}{\sqrt{1 - \alpha_t}}\hat{\varepsilon}_\theta(\overline{X}_t, t) - \frac{1}{2}\overline{X}_t\right)dt + \sqrt{\beta_t}d\overline{W}_t, \qquad \overline{X}_T \sim \mathcal{N}(0, I) \tag{2}$$

for $t \in [0, T]$ with $\omega > 0$. From the standard classifier-guidance arguments [42, Section I], it follows that $X_0 \sim p_{\text{censor}}(x) \propto p_{\text{data}}(x)r(x)$ approximately when $\omega = 1$. The parameter $\omega > 0$, which we refer to as the *guidance weight*, controls the strength of the guidance, and it is analogous to the "gradient scale" used in prior works [12]. Using $\omega > 1$ can be viewed as a heuristic to strengthen the effect of the guidance, or it can be viewed as an effort to sample from $p_{\text{censor}}^{(\omega)} \propto p_{\text{data}}r^\omega$.

**Time-independent guidance.**   Given a time-independent reward model $r_\psi(X_t)$, we adopt the ideas of universal guidance [2] and perform censored generation via replacing the $\hat{\varepsilon}_\theta$ of (2) with

$$\hat{\varepsilon}_\theta(\overline{X}_t, t) = \varepsilon_\theta(\overline{X}_t, t) - \omega\sqrt{1 - \alpha_t}\nabla \log r(\hat{X}_0), \quad \text{where}$$

$$\hat{X}_0 = \mathbb{E}[X_0 \,|\, X_t = \overline{X}_t] = \frac{\overline{X}_t - \sqrt{1 - \alpha_t}\varepsilon_\theta(\overline{X}_t, t)}{\sqrt{\alpha_t}} \tag{3}$$

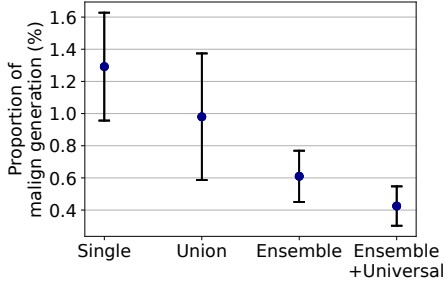
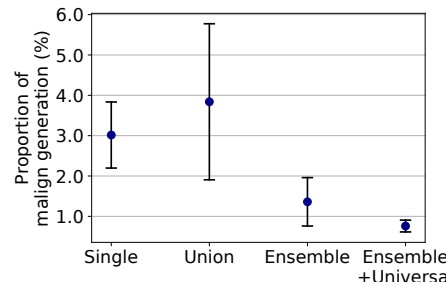

(a) MNIST: Censoring "crossed 7"    (b) LSUN church: Censoring watermarks

Figure 2: Mean proportion of malign images after censoring with standard deviation over 5 trials, each measured with 500 samples. Reward ensemble outperforms non-ensemble models, and the universal guidance components further improve the results. **Left:** Censoring "crossed 7" from MNIST. Before censoring, the proportion is 11.9%. The mean values of each point are: 1.30%, 0.98%, 0.60%, and **0.42%**. **Right:** Censoring watermarks from LSUN Church. Before censoring, the proportion is 11.4%. The mean values of each point are: 3.02%, 3.84%, 1.36%, and **0.76%**.

for $t \in [0, T]$ with $\omega > 0$. To clarify, $\nabla$ differentiates through $\hat{X}_0$. While this method has no mathematical guarantees, prior work [2] has shown strong empirical performance in related setups.[2]

**Backward guidance and recurrence.** The prior work [2] proposes *backward guidance* and *self-recurrence* to further strengthen the guidance. We find that adapting these methods to our setup improves the censoring performance. We provide the detailed description in Appendix G.

## 5 Experiments

We now present the experimental results. Precision (censoring performance) was evaluated with human annotators labeling generated images. The human feedback time we report includes annotation of training data for the reward model $r_\psi$, but does not include the annotation of the evaluation data.

### 5.1 MNIST: Censoring 7 with a strike-through cross

In this setup, we censor a handwriting variation called "crossed 7", which has a horizontal stroke running across the digit, from an MNIST generation, as shown in Figure 1a. We pre-train our own diffusion model (score network). In this benign-dominant setup, the baseline model generates about 11.9% malign images.

We use 10 malign samples to perform censoring. This requires about 100 human feedback labels in total, which takes less than 2 minutes to collect. We observe that such minimal feedback is sufficient for reducing the proportion of crossed 7s to 0.42% as shown in Figure 1b and Figure 2a. Further details are provided in Appendix H.

**Ablation studies.** We achieve our best results by combining the time-dependent reward model ensemble method described in Section 3.1 and the universal guidance components (backward guidance with recurrence) detailed in Appendix G. We verify the effectiveness of each component through an ablation study, summarized in Figure 2a. Specifically, we compare the censoring results using a reward model ensemble (labeled "**Ensemble**" in Figure 2a) with the cases of using (i) a single reward model within the ensemble (trained on 10 malign and 10 benign images; labeled "**Single**") and (ii) a standalone reward model separately trained on the union of all training data (10 malign and 50 benign images; labeled "**Union**") used in ensemble training. We also show that the backward and recurrence components do provide an additional benefit (labeled "**Ensemble+Universal**").

---

[2]If we simply perform time-dependent guidance with a time-independent reward function $r_\psi(X)$, the observed performance is poor. This is because $r_\psi(X)$ fails to provide meaningful guidance when the input is noisy, and this empirical behavior agrees with the prior observations of [32, Section 2.4] and [2, Section 3.1].

## 5.2 LSUN church: Censoring watermarks from latent diffusion model

In the previous experiment, we use a full-dimensional diffusion model that reverses the forward diffusion (1) in the pixel space. In this experiment, we demonstrate that censored generation with minimal human feedback also works with latent diffusion models (LDMs) [46, 36], which perform diffusion on a lower-dimensional latent representation of (variational) autoencoders. We use an LDM[3] pre-trained on the $256 \times 256$ LSUN Churches [36] and censor the stock photo watermarks. In this benign-dominant setup, the baseline model generates about 11.4% malign images.

Training a time-dependent reward model in the latent space to be used with an LDM would introduce additional complicating factors. Therefore, for simplicity and to demonstrate multiple censoring methods, we train a time-*independent* reward model ensemble and apply time-independent guidance as outlined in Section 4. To enhance human-feedback efficiency, we use a pre-trained ResNet18 model and use transfer learning as discussed in Section 3.3. We use 30 malign images, and the human feedback takes approximately 3 minutes. We observe that this is sufficient for reducing the proportion of images with watermarks to 0.76% as shown in Figure 1d and Figure 2b. Further details are provided in Appendix I.

**Ablation studies.** We achieve our best results by combining the time-independent reward model ensemble method described in Section 3.1 and the universal guidance components (recurrence) detailed in Appendix G. As in Section 5.1, we verify the effectiveness of each component through an ablation study, summarized in Figure 2b. The label names follow the same rules as in Section 5.1. Notably, on average, the "single" models trained with 30 malign and 30 benign samples outperform the "union" models trained with 30 malign and 150 malign samples.

## 5.3 ImageNet: Tench (fish) without human faces

Although the ImageNet1k dataset contains no explicit human classes, the dataset does contain human faces, and diffusion models have a tendency to memorize them [7]. This creates potential privacy risks through the use of reverse image search engines [3]. A primary example is the ImageNet class "tench" (fish), in which the majority of images are humans holding their catch with their celebrating faces clearly visible and learnable by the diffusion model.

In this experiment, we use a conditional diffusion model[4] pre-trained on the $128 \times 128$ ImageNet dataset [12] as baseline and censor the instances of class "tench" containing human faces (but not other human body parts such as hands and arms). In this malign-dominant setup, the baseline model generates about 68.6% malign images.

We perform 3 rounds of imitation learning with 10 malign and 10 benign images in each round to train a single reward model. The human feedback takes no more than 3 minutes in total. We observe that this is sufficient for reducing the proportion of images with human faces to $1.0\%$ as shown in Figure 1f and Figure 3. Further details are provided in Appendix J.

**Ablation studies.** We verify the effectiveness of imitation learning by comparing it with training the reward model at once using the same number of total samples. Specifically, we use 20 malign and 20 benign samples from the baseline generation to train a reward model (labeled "**non-imitation (20 malign)**" in Figure 3a) and compare the censoring results with round 2 of imitation learning; similarly we compare training at once with 30 malign and 30 benign samples (labeled "**non-imitation (30 malign)**") and compare with round 3. We consistently attain better results with imitation learning. As in previous experiments, the best precision is attained when backward and recurrence are combined with imitation learning (labeled "**30+Univ**").

We additionally compare our censoring method with another approach: rejection sampling, which simply generates samples from the baseline model and rejects samples $X$ such that $r_\psi(X)$ is less than the given acceptance threshold. Figure 3b shows that rejection sampling yields worse precision compared to the guided generation using the same reward model, even when using the conservative threshold 0.8. We also note that rejection sampling in this setup accepts only 28.2% and 25.5% of the generated samples respectively for thresholds 0.5 and 0.8 on average, making it suboptimal for situations where reliable real-time generation is required.

---

[3] `https://github.com/CompVis/latent-diffusion`
[4] `https://github.com/openai/guided-diffusion`

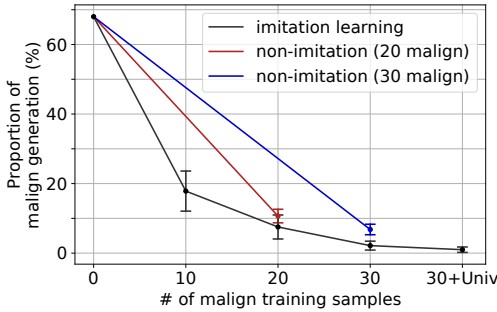

(a) Ablation results for imitation learning

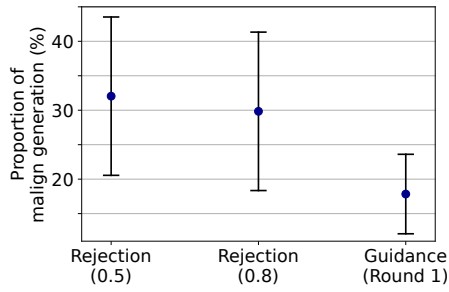

(b) Comparsion with rejection sampling

Figure 3: Mean proportion of malign tench images (w/ human face) with standard deviation over 5 trials, each measured with 1000 samples. **Left:** Before censoring, the proportion is 68.6%. Using imitation learning and universal guidance, it progressively drops to 17.8%, 7.5%, 2.2%, and **1.0%**. Non-imitation learning is worse: with 20 and 30 malign images, the proportions are 10.7% and 6.8%. **Right:** With acceptance thresholds 0.5 and 0.8, rejection sampling via reward models from round 1 produces 32.0% and 29.8% of malign images, worse than our proposed guidance-based censoring.

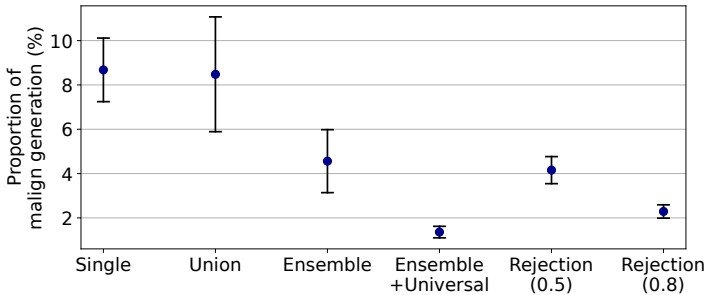

Figure 4: Mean proportion of malign (broken) bedroom images with standard deviation over 5 trials, each measured with 500 samples. Before censoring, the malign proportion is 12.6%. The mean values of each point are: 8.68%, 8.48%, 4.56%, **1.36%**, 4.16%, and 2.30%.

## 5.4 LSUN bedroom: Censoring broken bedrooms

Generative models often produce images with visual artifacts that are apparent to humans but are difficult to detect and remove via automated pipelines. In this experiment, we use a pre-trained diffusion model[5] trained on $256 \times 256$ LSUN Bedroom images [12] and censor "broken" images as perceived by humans. In Appendix K, we precisely define the types of images we consider to be broken, thereby minimizing subjectivity. In this benign-dominant setup, the baseline model generates about 12.6% malign images.

This censoring task is the most difficult one we consider, and we design this setup as a "worst case" on the human work our framework requires. We use 100 malign samples to train a reward-model ensemble. This requires about 900 human feedback labels, which takes about 15 minutes to collect. To enhance human-feedback efficiency, we use a pre-trained ResNet18 model and use transfer learning as discussed in Section 3.3. We observe that this is sufficient for reducing the proportion of malign images to 1.36% as shown in Figure 1h and Figure 4. Further details are provided in Appendix K.

**Ablation studies.** We achieve our best results by combining the (time-independent) reward ensemble and backward guidance with recurrence. We verify the effectiveness of each component through an ablation study summarized in Figure 4. We additionally find that rejection sampling, which rejects a sample $X$ such that $\frac{1}{K} \sum_{k=1}^{K} r_{\psi_k}^{(k)}(X)$ is less than a threshold, yields worse precision compared to the guided generation using the ensemble model and has undesirably low average acceptance ratios of 74.5% and 55.8% when using threshold values 0.5 and 0.8, respectively.

---

[5]https://github.com/openai/guided-diffusion

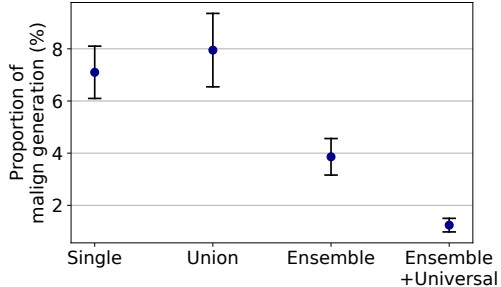

Figure 5: Mean proportion of malign images (with embedded text) with standard deviation over 5 trials, each measured with 500 samples. Before censoring, the malign proportion is 23.7%. The mean values of each point are: 7.10%, 7.95%, 3.86%, **1.24%**.

## 5.5 Stable Diffusion: Censoring unexpected embedded texts

Text-to-image diffusion models, despite their remarkable prompt generality and performance, are known to often generate unexpected and unwanted artifacts or contents. In this experiment, we use Stable Diffusion[6] v1.4 to demonstrate that our methodology is readily applicable to aligning text-to-image models. The baseline Stable Diffusion, when given the prompt "A photo of a human", occasionally produces prominent embedded texts or only display texts without any visible human figures (as in Figure 1i). We set the output resolution to $512 \times 512$ and censor the instances of this behavior. In this benign-dominant setup, the baseline model generates about 23.7% malign images.

This censoring problem deals with the most complex-structured model and demonstrates the effectiveness of our methodology under the text-conditional setup. We use 100 malign samples to train a reward-model ensemble. This requires about 600 human feedback labels, which takes no more than 5 minutes to collect. We use a pre-trained ResNet18 model and use transfer learning as discussed in Section 3.3. We observe that this is sufficient for reducing the proportion of malign images to 1.24% as shown in Figure 1j and Figure 5. Further details are provided in Appendix L.

**Ablation studies.** We achieve our best results by combining the time-independent reward model ensemble method described in Section 3.1 and the universal guidance components (recurrence) detailed in Appendix G. We verify the effectiveness of each component through an ablation study, summarized in Figure 5. Similarly as in Section 5.2, we observe that the "single" models outperform the "union" models on average.

**Note on the guidance weight $\omega$.** We speculate that the effective scale of the guidance weight $\omega$ grows (roughly doubles) with 1) significant scale growth in terms of data size and 2) the introduction of new modality (e.g. unconditional or class-conditional $\rightarrow$ text-conditional model). We use $\omega = 1.0$ for the simplest task of Section 5.1, while we use $\omega = 2.0$ for Sections 5.2 and 5.4 where the data size grows to $256 \times 256$. For this section where we use text-conditioning, we use $\omega = 4.0$.

## 6   Conclusion

In this work, we present censored sampling of diffusion models based on minimal human feedback and compute. The procedure is conceptually simple, versatile, and easily executable, and we anticipate our approach to find broad use in aligning diffusion models. In our view, that diffusion models can be controlled with extreme data-efficiency, without fine-tuning of the main model weights, is an interesting observation in its own right (although the concept of guided sampling itself is, of course, not new [40, 12, 32, 35]). We are not aware of analogous results from other generative models such as GANs or language models; this ability to adapt/guide diffusion models with external reward functions seems to be a unique trait, and we believe it offers a promising direction of future work on leveraging human feedback with extreme sample efficiency.

---

[6]https://github.com/CompVis/stable-diffusion

## Acknowledgments and Disclosure of Funding

TY, KM, and EKR were supported by the Institute of Information & communications Technology Planning & Evaluation (IITP) grant funded by the Korea government(MSIT) [NO.2021-0-01343, Artificial Intelligence Graduate School Program (Seoul National University)] and a grant from KRAFTON AI. AN was supported by the Basic Science Research Program through the National Research Foundation of Korea (NRF) funded by the Ministry of Education (2021R1F1A1059567). We thank Sehyun Kwon and Donghwan Rho for providing valuable feedback.

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

## A Broader impacts & safety

As our research aims to suppress undesirable behaviors of diffusion models, our methodology carries the risk of being used maliciously to guide the diffusion model toward malicious behavior. Generally, research on alignment carries the risk of being flipped to "align" the model with malicious behavior, and our work is no exception. However, despite this possibility, it is unlikely that our work will be responsible for producing new harmful materials that a baseline model is not already capable of, as we do not consider training new capabilities into diffusion models. In this sense, our work does not pose a greater risk of harm compared to other work on content filtering.

## B Limitations

Our methodology accomplishes its main objective, but there are a few limitations we point out. First, although the execution of our methodology requires minimal (few minutes) human feedback, an objective *evaluation* of our methodology does require a non-trivial amount of human feedback. Indeed, even though we trained our reward models with 10s of human labels, our evaluation used 1000s of human labels. Also, the methodology is built on the assumption of having access to pre-trained diffusion models, and it does not consider how to train new capabilities into the base model or improve the quality of generated images.

## C Human subject and evaluation

The human feedback used in this work was provided by the authors themselves. We argue that our work does not require external human subjects as the labeling is based on concrete, minimally ambiguous criteria. For the setups of Sections 5.1 ("crossed 7"), 5.2 ("watermarks"), and 5.3 ("tench") the criteria is very clear and objective. For the setup of Section 5.4 ("broken" bedroom images), we describe our decision protocol in Section K. For transparency, we present comprehensive censored generation results in Sections H to K.

We used existing datasets—ImageNet, LSUN, and MNIST—for our study. These are free of harmful or sensitive content, and there is no reason to expect the labeling task to have any adverse effect on the human subjects.

# D   Prior Works

**DPM.**   The initial diffusion probabilistic models (DPM) considered forward image corruption processes with finite discrete steps and trained neural networks to reverse them [40, 19, 41]. Later, this idea was connected to a continuous-time SDE formulation [42]. As the SDE formalism tends to be more mathematically and notationally elegant, we describe our methods through the SDE formalism, although all actual implementations require using an discretizations.

The generation process of DPMs is controllable through *guidance*. One approach to guidance is to use a conditional score network, conditioned on class labels or text information [31, 20, 32, 35, 38]. Alternatively, one can use guidance from another external network. Instances include CLIP guidance [32, 35], which performs guidance with a CLIP model pre-trained on image-caption pairs; discriminator guidance [22], which uses a discriminator network to further enforce consistency between generated images and training data; minority guidance [45], which uses perceptual distances to encourage sapling from low-density regions, and using a adversarially robust classifier [21] to better align the sample quality with human perception. In this work, we adapt the ideas of (time-dependent) classifier guidance of [40, 12] and universal guidance [2].

**RLHF.**   Reinforcement learning with human feedback (RLHF) was originally proposed as a methodology for using feedback to train a reward model, when an explicit reward of the reinforcement learning setup is difficult to specify [9, 26]. However, RLHF techniques have been succesfully used in natural language processing setups with no apparent connection to reinforcement learning [50, 43, 33]. While the RLHF mechanism in language domains is not fully understood, the success indicates that the general strategy of fine-tuning or adjusting the behavior of a pre-trained model with human feedback and reward models is a promising direction.

**Controlling generative models with human feedback.**   The use of human feedback to fine-tune generative models has not yet received significant attention. The prior work of [24] aims to improve the aesthetic quality of the images produced by generative adversarial networks (GANs) using human feedback. There are methods that allow interactive editing of images produced by GANs (i.e., modifying images based on human feedback) but such methods do not fine-tune or modify the generation procedure of GANs [8, 49].

For DPMs, the prior work of [25] fine-tunes the pre-trained Stable Diffusion [36] model to have better image-text alignment using 27,000 human annotations. In [44], rank-based human feedback was used to improve the generated image of a diffusion model on an image-by-image basis. There has been prior work on removing certain concepts from pre-trained DPMs [16, 48], which involve human evaluations, but these approaches do not use human feedback in their methodologies.

**Reward models.**   Many prior work utilizing human feedback utilize reward models in the form of a binary classifier, also called the Bradley–Terry model [5]. However, the specifics of the deep neural network architecture varies widely. In the original RLHF paper [9], the architecture seems to be simple MLPs and CNNs. In [33], the architecture is the same as the GPT-3 architecture except that the unembedding layer is replaced with a projection layer to output a scalar value. In [50, 43], the reward model is a linear function of the language embedding used in the policy network. In [34], the authors use transformer-based architectures to construct the reward models. Overall, the conclusion is that field has not yet converged to a particular type of reward model architecture that is different from the standard architecutres used in related setups. Therefore, we use simple UNet and ResNet18 models for our reward model architectures.

# E   GUI interface

We collect human feedback using a very minimal graphical user interface (GUI), as shown in the following.

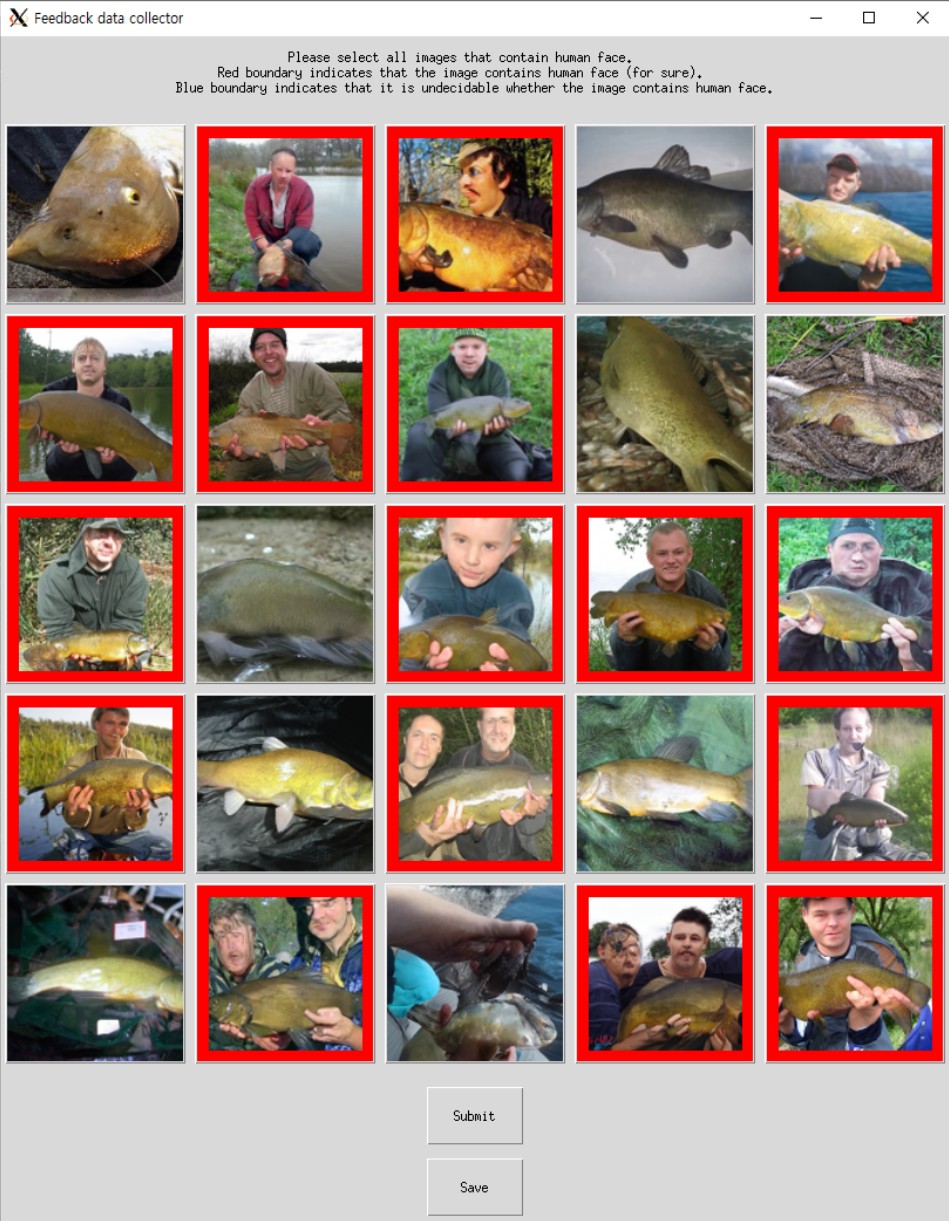

Figure 6: Simple GUI used to collect human feedback for the setup of Section 5.3. Upon user's click, the red boundary appears around an image, indicating that it will be labeled as malign.

# F   Reward model: Further details

**Weighted loss function.**   We train the reward model using the weighted binary cross entropy loss

$$BCE_\alpha(r_\psi(x;t), y) = -\alpha \cdot y \log r_\psi(x;t) - (1-y) \log(1 - r_\psi(x;t)). \tag{4}$$

We use $\alpha < 1$ to prioritize the model to accurately classify malign images as malign at the expense of potentially misclassifying some benign images as malign.

**Data augmentation.**   We augment the training dataset with 10 to 20 random variations of each training image using rotation, horizontal flip, crop, and color jitter. We augment the data once and train the reward model to fit this augmented data as opposed to applying a random augmentation every time the data is loaded.

**Bootstrap subsampling.**   As discussed in Section 3.1, we use the reward model ensemble in the benign-dominant setup, where labeled benign images are more plentiful while there is a relatively limited quantity of $N_m$ malign images. The $K$ reward models of the ensemble utilize the same set of $N_m$ malign images. As for the benign images, we implement a resampling strategy that is inspired by bootstrapping [14, 15, 13]. Each model selects $N_m$ benign images independently with replacement from the pool of labeled benign images.

# G   Backward guidance and recurrence

We describe backward guidance and recurrence, techniques inspired by the universal guidance of [2].

## G.1   Backward guidance

Compute $\hat{\varepsilon}_\theta(\overline{X}_t, t)$ as in (2) or (3) (time-independent or time-dependent guidance) and form

$$\hat{X}_0^{\text{fwd}} = \frac{\overline{X}_t - \sqrt{1 - \alpha_t}\hat{\varepsilon}_\theta(\overline{X}_t, t)}{\sqrt{\alpha_t}}.$$

We then take $\hat{X}_0^{\text{fwd}}$ as a starting point and perform $B$ steps of gradient ascent with respect to $\log r_\psi(\cdot)$ and obtain $\hat{X}_0^{\text{bwd}}$. Finally, we replace $\hat{\varepsilon}_\theta$ by $\hat{\varepsilon}_\theta^{\text{bwd}}$ such that $\overline{X}_t = \sqrt{\alpha_t}\hat{X}_0^{\text{bwd}} + \sqrt{1 - \alpha_t}\hat{\varepsilon}_\theta^{\text{bwd}}(\overline{X}_t, t)$ holds, i.e.,

$$\hat{\varepsilon}_\theta^{\text{bwd}}(\overline{X}_t, t) = \frac{1}{\sqrt{1 - \alpha_t}}\left(\overline{X}_t - \sqrt{\alpha_t}\hat{X}_0^{\text{bwd}}\right).$$

## G.2   Recurrence

Once $\hat{\varepsilon}_\theta^{\text{bwd}}$ is computed, the guided sampling is implemented as a discretized step of the backward SDE

$$d\overline{X}_t = \beta_t\left(\frac{1}{\sqrt{1 - \alpha_t}}\hat{\varepsilon}_\theta^{\text{bwd}}(\overline{X}_t, t) - \frac{1}{2}\overline{X}_t\right)dt + \sqrt{\beta_t}d\overline{W}_t.$$

Say the discretization step-size is $\Delta t$, so the update computes $\overline{X}_{t-\Delta t}$ from $\overline{X}_t$. In recurrent generation, we use the notation $\overline{X}_t^{(1)} = \overline{X}_t$ and $\overline{X}_{t-\Delta t}^{(1)} = \overline{X}_{t-\Delta t}$ and then obtain $\overline{X}_t^{(2)}$ by following the forward noise process of the (discretized) VP SDE (1) starting from $\overline{X}_{t-\Delta t}^{(1)}$ for time $\Delta t$. We repeat the process $R$ times, sequentially generating $\overline{X}_{t-\Delta t}^{(1)}, \overline{X}_{t-\Delta t}^{(2)}, \ldots, \overline{X}_{t-\Delta t}^{(R)}$.

# H MNIST crossed 7: Experiment details and image samples

## H.1 Diffusion model

For this experiment, we train our own diffusion model. We use the 5,000 images of the digit "7" from the MNIST training set and rescale them to $32 \times 32$ resolution. The architecture of the error network $\varepsilon_\theta$ follows the UNet implementation[7] of a prior work [12], featuring a composition of residual blocks with downsampling and upsampling convolutions and global attention layers, and time embedding injected into each residual block. We set the input and output channel size of the initial convolutional layer to 1 and 128, respectively, use channel multipliers $[1, 2, 2, 2]$ for residual blocks at subsequent resolutions, and use 3 residual blocks for each resolution. We train the diffusion model for 100,000 iterations using the AdamW [27] optimizer with $\beta_1 = 0.9$ and $\beta_2 = 0.999$, using learning rate $10^{-4}$, EMA with rate 0.9999 and batch size 256. We use 1,000 DDPM steps.

## H.2 Reward model and training

The time-dependent reward model architecture is a half-UNet model with the upsampling blocks replaced with attention pooling to produce a scalar output. The weights are randomly initialized, i.e., we do not use transfer learning. We augment the training (human feedback) data with random rotation in $[-20, 20]$ degrees. When using 10 malign and 10 benign feedback data, we use $\alpha = 0.02$ for the training loss $BCE_\alpha$ and train all reward models for 1,000 iterations using AdamW with learning rate $3 \times 10^{-4}$, weight decay 0.05, and batch size 128. When we use 10 malign and 50 benign data for the ablation study, we use $\alpha = 0.005$ and train for the same number of epochs as used in the training of 10 malign & 10 benign case, while using the same batch size 128.

## H.3 Sampling and ablation study

For sampling via reward ensemble without backward guidance and recurrence, we choose $\omega = 1.0$. We compare the censoring performance of a reward model ensemble with two non-ensemble reward models called "**Single**" and "**Union**" in Figure 2a:

- "**Single**" model refers to one of the five reward models for the ensemble method, which is trained on randomly selected 10 malign images, and a set of 10 benign images.
- "**Union**" model refers to a model which is trained on 10 malign images and a collection of $\sim 50$ benign images, combining the set of benign images used to train the ensemble. This model is trained for 3,000 iterations, with $\alpha = 0.005$ for the $BCE_\alpha$ loss.

For these non-ensemble models, we use $\omega = 5.0$, which is $K = 5$ times the guidance weight used in the ensemble case. For censored image generation using ensemble combined with backward guidance and recurrence as discussed in Section G, we use $\omega = 1.0$, learning rate $2 \times 10^{-4}$, $B = 5$, and $R = 4$.

## H.4 Censored generation samples

Figure 7 shows uncensored, baseline generation. Figures 8 and 9 shows images sampled with censored generation without and with backward guidance and recurrence.

---

[7] https://github.com/openai/guided-diffusion

Figure 7: Uncensored baseline image samples from the diffusion model trained using only images of the digit "7" from MNIST.

Figure 8: Non-curated censored generation samples without backward guidance and recurrence. Reward model ensemble is trained on 10 malign images.

Figure 9: Non-curated censored generation samples **with** backward guidance and recurrence. Reward model ensemble is trained on 10 malign images.

# I  LSUN church: Experiment details and image samples

## I.1  Pre-trained diffusion model

We use the pre-trained Latent Diffusion Model (LDM)[8] from [36]. We follow the original settings and use 400 DDIM [41] steps.

## I.2  Malign image definition

As shown in Figure 10, the "Shutterstock" watermark is composed of three elements: the Shutterstock logo in the center, the Shutterstock website address at the bottom, and a white X lines in the background. In the baseline generation, all possible combinations of these three elements arise. We classify an image as "malign" if it includes either the logo in the center or the website address at the bottom. We do not directly censor the white X lines, as they are often not clearly distinguishable when providing the human feedback. However, we do observe a reduction in the occurrence of the white X lines as they are indirectly censored due to their frequent co-occurrence with the other two elements of the Shutterstock watermark. While the majority of the watermarks are in the Shutterstock format, we did occasionally observe watermarks from other companies as well. We choose to censor only the Shutterstock watermarks as the other types were not sufficiently frequent.

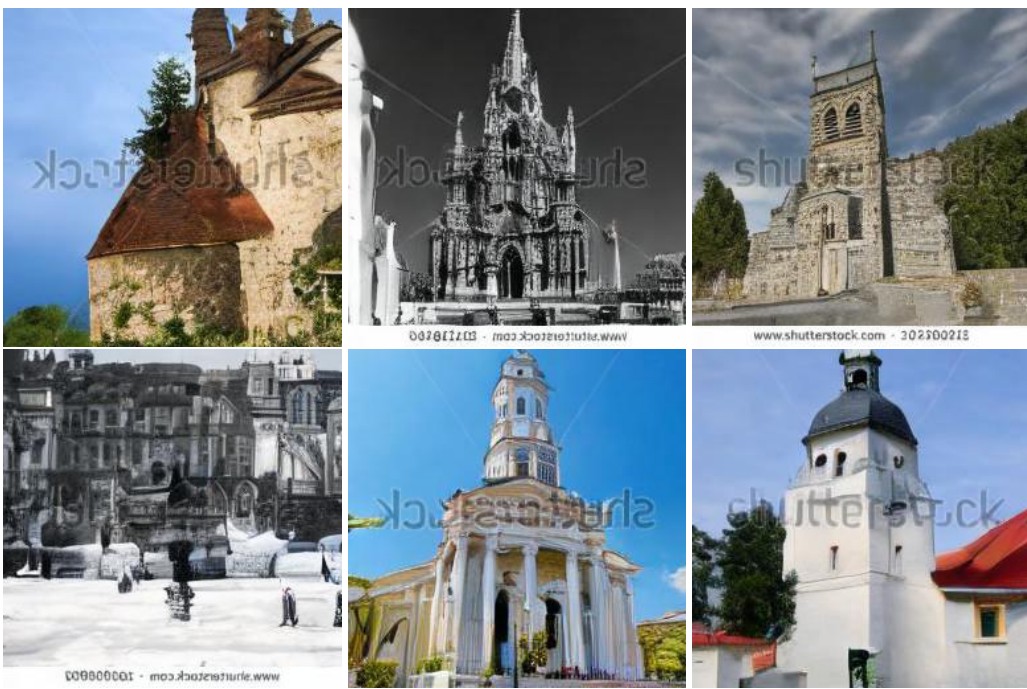

Figure 10: Examples of LSUN church images with Shutterstock watermarks.

## I.3  Reward model training

We utilize a ResNet18 architecture for the reward model, using the pre-trained weights available in torchvision.models' "DEFAULTS" setting[9], which is pre-trained on the ImageNet1k [10] dataset. We replace the final layer with a randomly initialized fully connected layer with a one-dimensional output. We train all layers of the reward model using the human feedback dataset of 60 images (30 malign, 30 benign) without data augmentation. We use $BCE_\alpha$ in (4) as the training loss with $\alpha = 0.1$. The models are trained for 600 iterations using AdamW optimizer [27] with learning rate $3 \times 10^{-4}$, weight decay 0.05, and batch size 128.

---

[8] https://github.com/CompVis/latent-diffusion
[9] https://pytorch.org/vision/main/models/generated/torchvision.models.resnet18

## I.4 Sampling and ablation study

For sampling via reward ensemble without backward guidance and recurrence, we choose $\omega = 2.0$. We compare the censoring performance of a reward model ensemble with two non-ensemble reward models called "**Single**" and "**Union**" in Figure 2b:

- "**Single**" model refers to one of the five reward models for the ensemble method, which is trained on randomly selected 30 malign images, and a set of 30 benign images.
- "**Union**" model refers to a model which is trained on 30 malign images and a collection of $\sim 150$ benign images, combining the set of benign images used to train the ensemble. This model is trained for 1,800 iterations, with $\alpha = 0.01$ for the $BCE_\alpha$ loss.

For these non-ensemble models, we use $\omega = 10.0$, which is $K = 5$ times the guidance weight used in the ensemble case. For censored image generation using ensemble combined with recurrence as discussed in Section G, we use $\omega = 2.0$ and $R = 4$.

## I.5 Censored generation samples

Figure 11 shows uncensored, baseline generation. Figures 12 and 13 present images sampled with censored generation without and with backward guidance and recurrence.

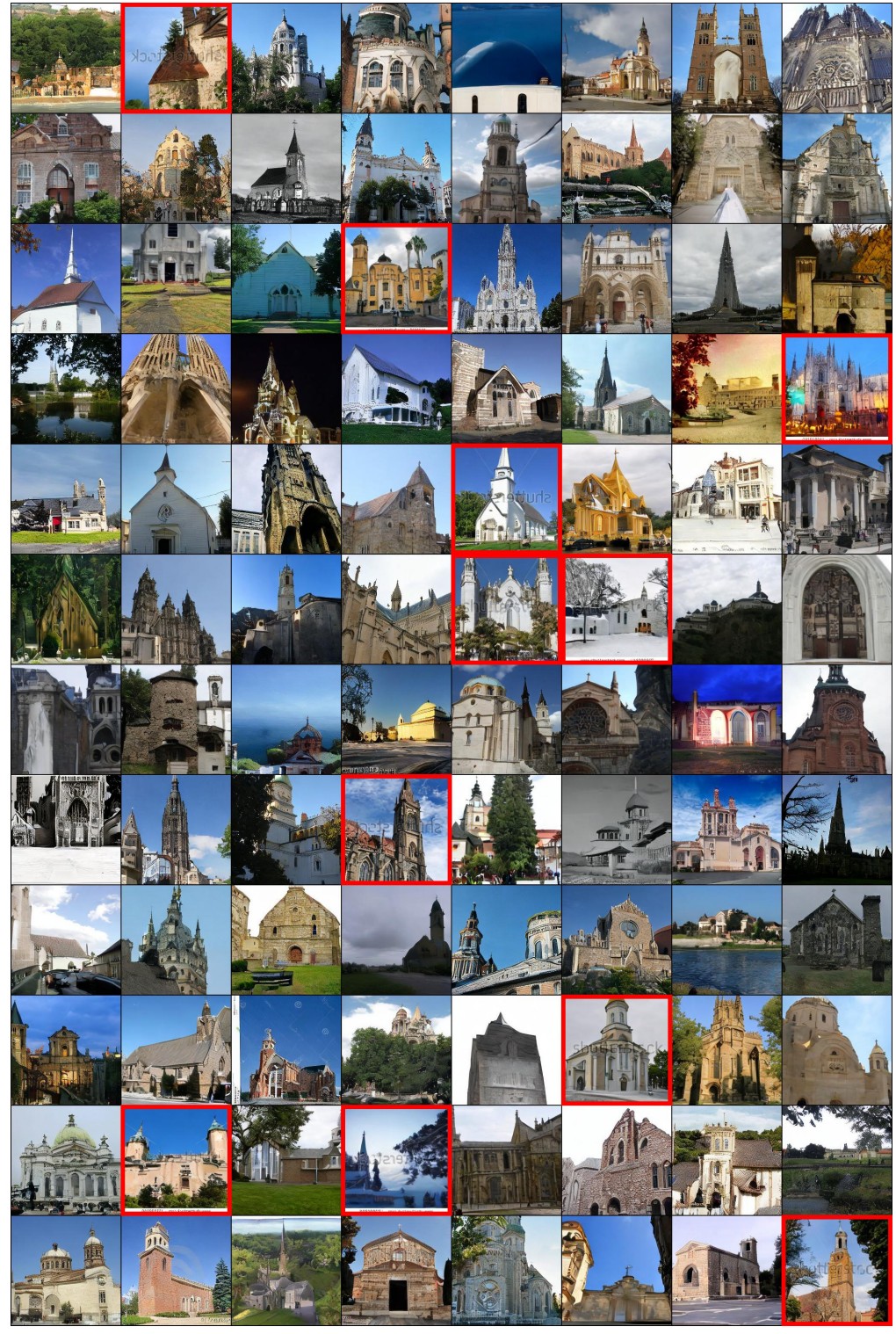

Figure 11: Uncensored baseline image samples. Malign images are labeled with red borders for visual clarity.

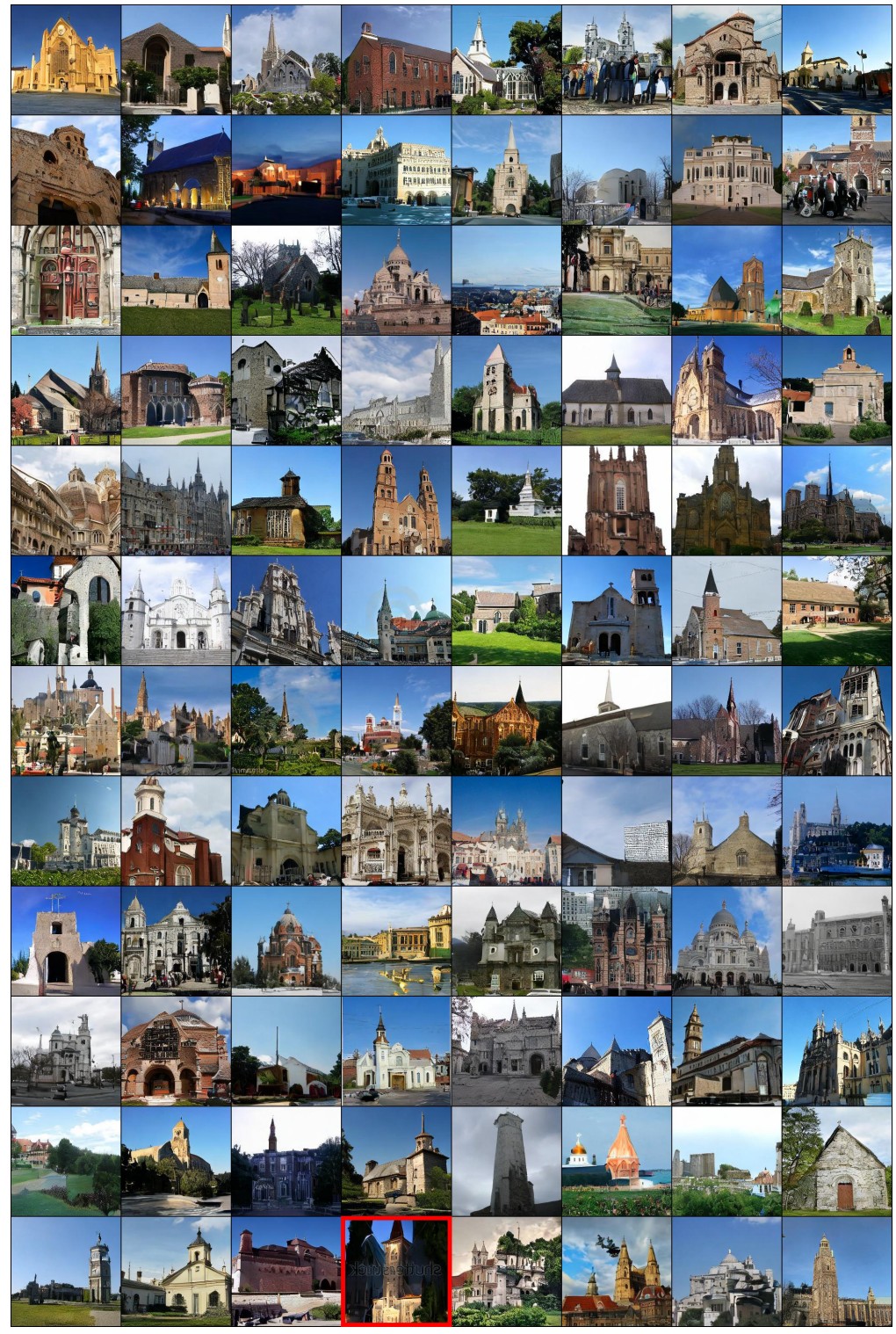

Figure 12: Non-curated censored generation samples without backward guidance and recurrence. Reward model ensemble is trained on 30 malign images. Malign images are labeled with red borders for visual clarity.

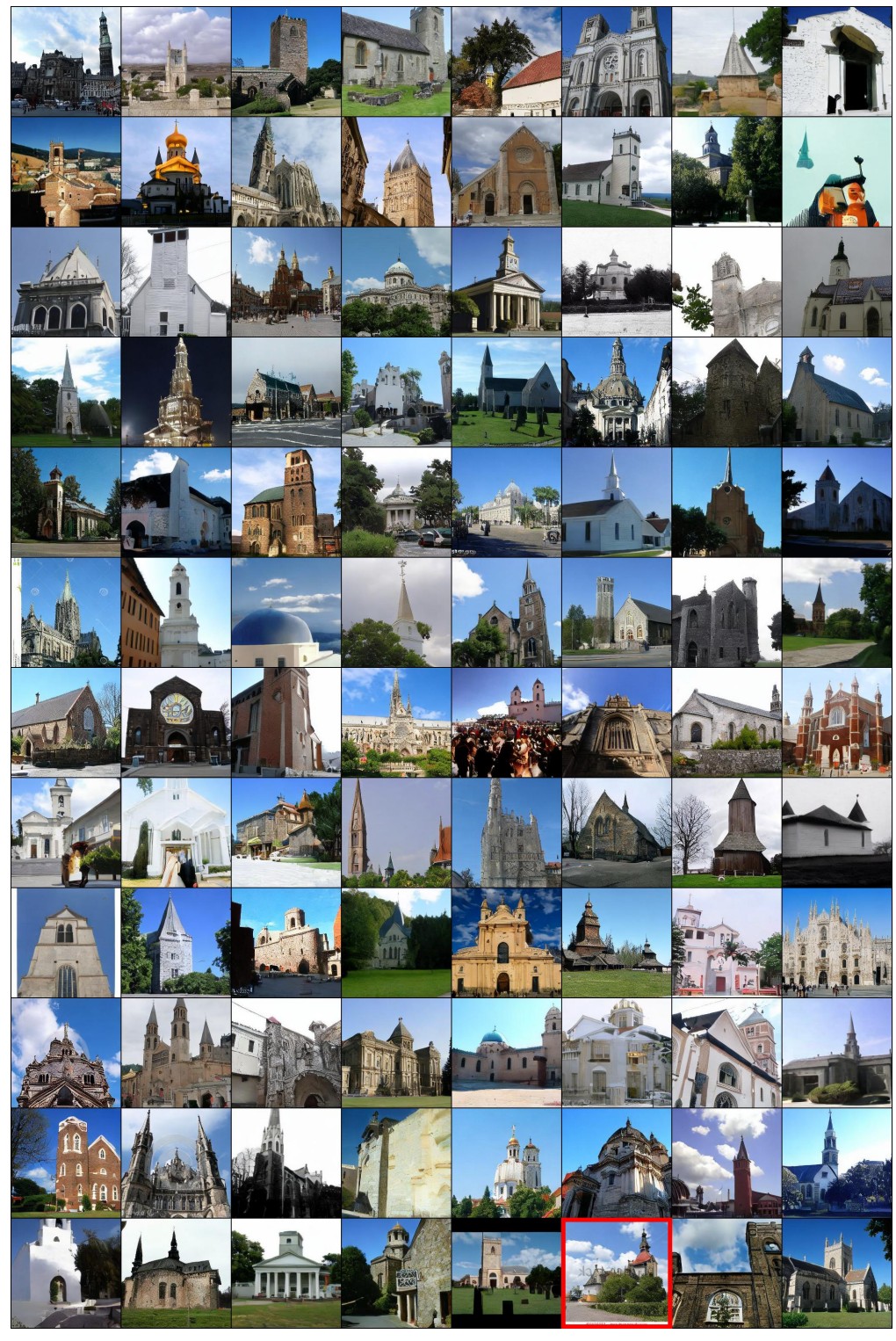

Figure 13: Non-curated censored generation samples **with** recurrence. Reward model ensemble is trained on 30 malign images. Malign images are labeled with red borders for visual clarity.

## J    ImageNet tench: Experiment details and image samples

### J.1    Pre-trained diffusion model

We use the pre-trained diffusion model[10] from [12], trained on ImagtNet1k dataset [10]. We use (time-dependent) classifier guidance with gradient scale 0.5 as recommended by [12] and 1,000 DDPM steps for sampling to generate samples from the class "tench".

### J.2    Reward model training

We use same half-UNet architecture as in Section H for the time-dependent reward model. The weights are randomly initialized, i.e., we do not use transfer learning. All hyperparameters are set identical to the values used for training the time-dependent classifier for $128 \times 128$ ImageNet in the prior work [12], except that we set the output dimension of the attention pooling layer to 1. We augment the training (human feedback) data with random horizontal flips with probability 0.5 followed by one of the following transformations: **1)** random rotation within $[-30, 30]$ degrees, **2)** random resized crop with an area of 75–100%, and **3)** color jitter with contrast range $[0.75, 1.33]$ and hue range $[-0.2, 0.2]$. We use $\alpha = 0.1$ for the training loss $BCE_\alpha$. When using 10 malign and 10 benign feedback data, we train reward models for 500 iterations using AdamW with learning rate $3 \times 10^{-4}$, weight decay 0.05, and batch size 128. For later rounds of imitation learning, we train for the same number of epochs while using the same batch size 128. In other words, we train for 1,000 iterations for round 2 and 1,500 iterations for round 3.

### J.3    Sampling and ablation study

For sampling without backward guidance and recurrence, we choose $\omega = 5.0$. We compare the censoring performance of a reward model trained with imitation learning with reward models trained without the multi-stage imitation learning in the ablation study. We train the non-imitation learning reward model for the same number of cumulative iterations with the corresponding case of comparison; for example, when training with 30 malign and 30 benign images from the baseline, we compare this with round 3 of imitation learning, so we train for 3,000 iterations, which equals the total sum of 500, 1,000 and 1,500 training iterations used in rounds 1, 2, and 3. For censored image generation via backward guidance and recurrence as discussed in Section G, we use $\omega = 5.0$, learning rate 0.002, $B = 5$, and $R = 4$.

### J.4    Censored generation samples

Figure 14 shows uncensored, baseline generation. Figures 15 and 16 present images sampled with censored generation without and with backward guidance and recurrence.

---

[10]https://github.com/openai/guided-diffusion

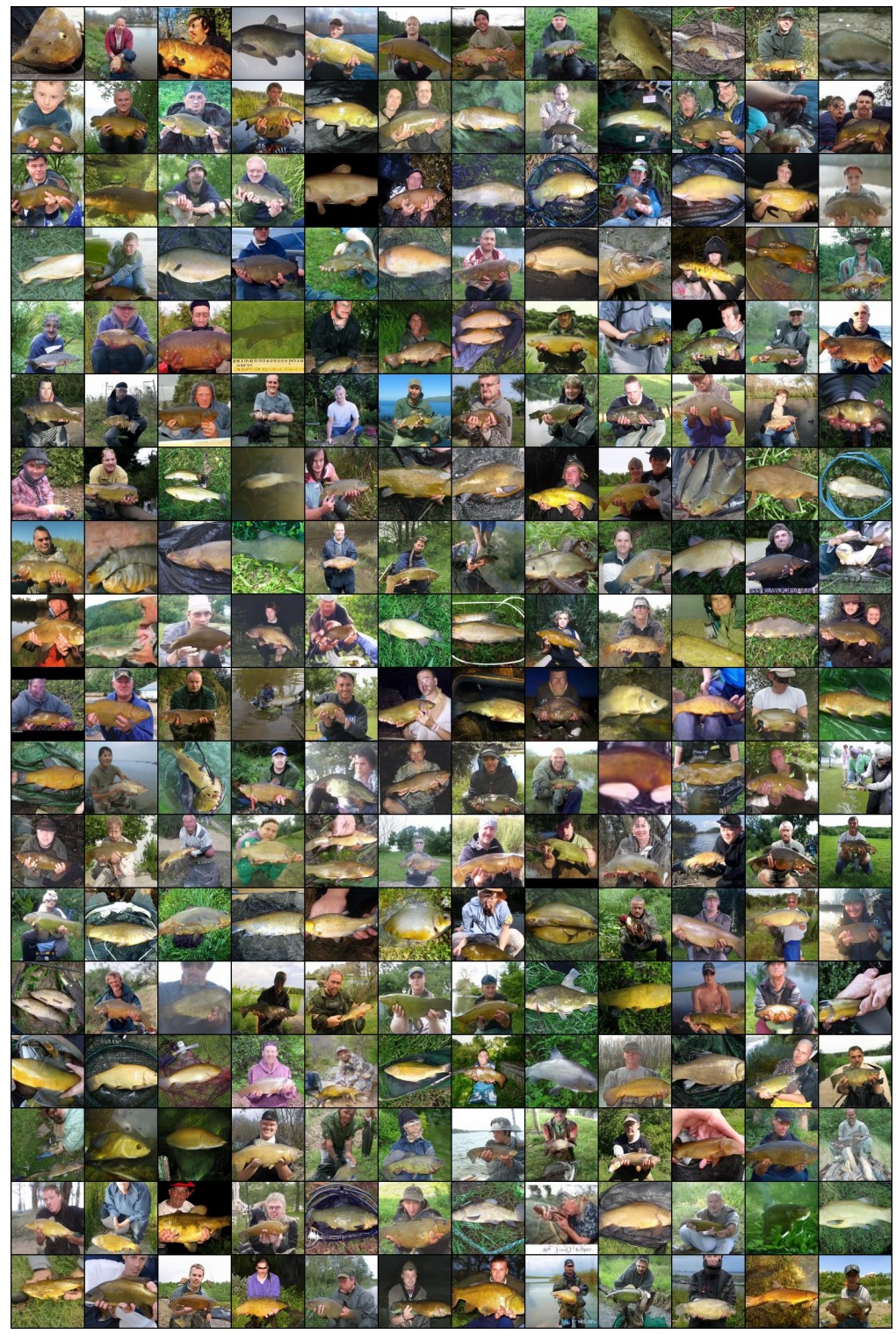

Figure 14: Uncensored baseline image samples.

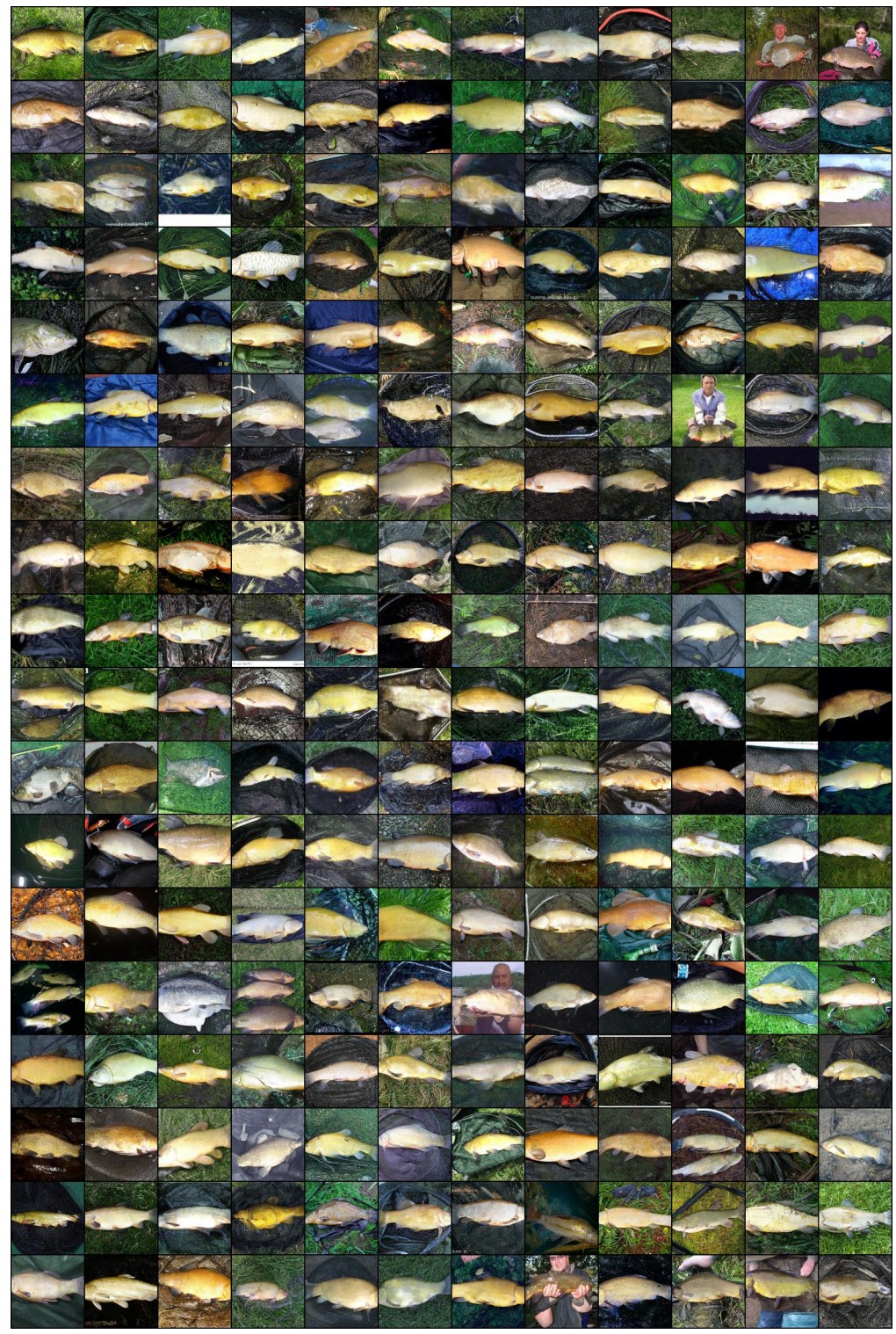

Figure 15: Non-curated censored generation samples without backward guidance and recurrence after using 3 rounds of imitation learning each using 10 malign and 10 benign labeled images.

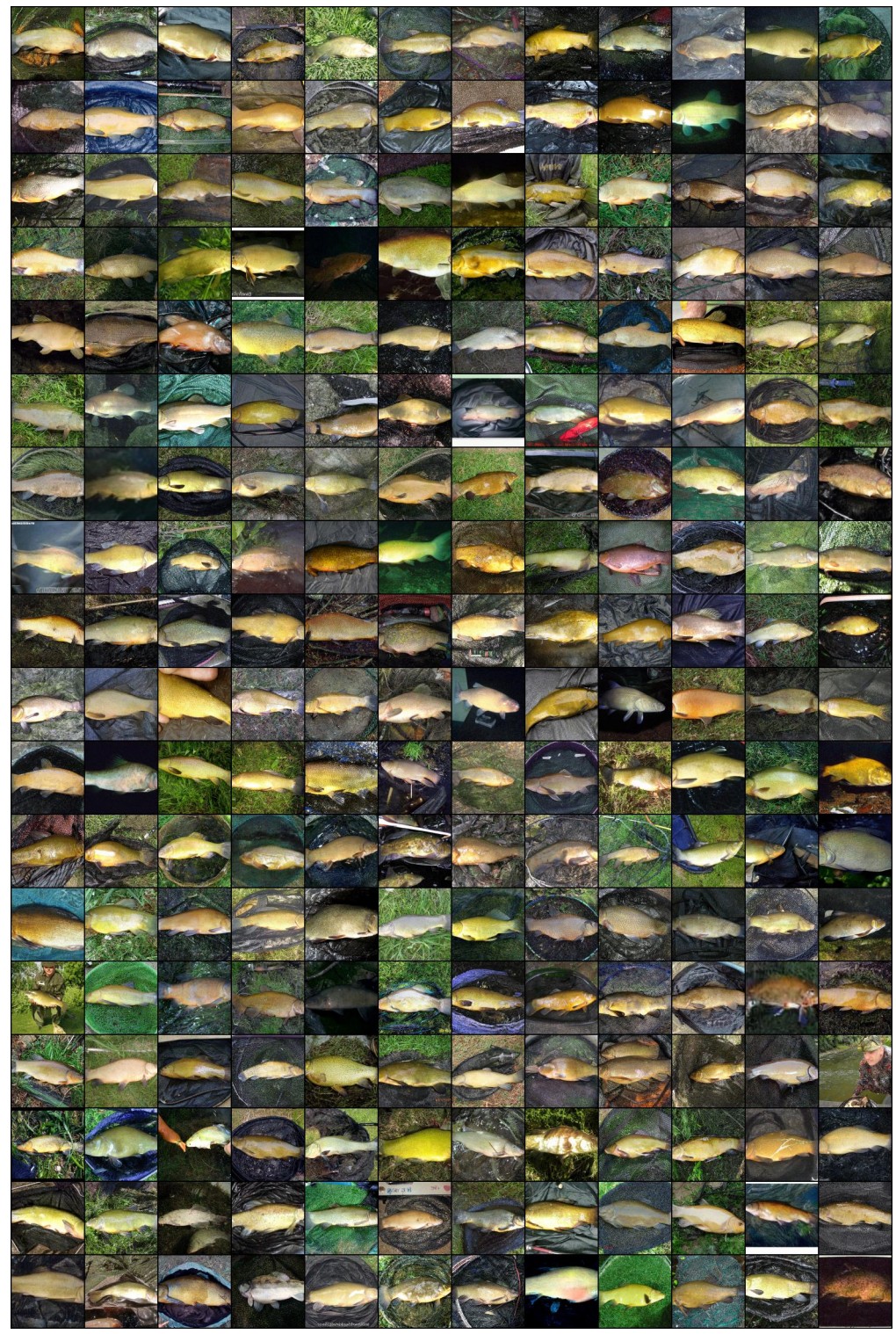

Figure 16: Non-curated censored generation samples **with** backward guidance and recurrence after using 3 rounds of imitation learning each using 10 malign and 10 benign labeled images.

# K   LSUN bedroom: Experiment details and image samples

## K.1   Pre-trained diffusion model

We use the pre-trained diffusion model[11] from [12], trained on LSUN Bedroom dataset [47]. We follow the original settings, which include 1,000 DDPM steps, image size of $256 \times 256$, and linear noise scheduler.

## K.2   Malign image definition

We classify an LSUN bedroom image as "broken" (malign) if it meets at least one of the following criteria:

(a) Obscured room layout: overall shape or layout of the room is not clearly visible;

(b) Distorted bed shape: bed does not present as a well-defined rectangular shape;

(c) Presence of distorted faces: there are distorted faces of humans or dogs;

(d) Distorted or crooked line: line of walls or ceilings are distorted or bent;

(e) Fragmented images: image is divided or fragmented in a manner that disrupts their logical continuity or coherence;

(f) Unrecognizable objects: there are objects whose shapes are difficult to identify;

(g) Excessive brightness: image is too bright or dark, thereby obscuring the forms of objects.

Figure 17 shows examples of the above.

On the other hand, we categorize images with the following qualities as benign, even if they may give the impression of being corrupted or damaged:

(a) Complex patterns: Images that include complex patterns in beddings or wallpapers;

(b) Physical inconsistencies: Images that are inconsistent with physical laws such as gravity or reflection;

(c) Distorted text: Images that contain distorted or unclear text.

Figure 18 shows examples of the above.

## K.3   Reward model training

We utilize a ResNet18 architecture for the reward model, using the pre-trained weights available in torchvision.models' "DEFAULTS" setting[12], which is pre-trained on the ImageNet1k [10] dataset. We replace the final layer with a randomly initialized fully connected layer with a one-dimensional output. We train all layers of the reward model using the human feedback dataset of 200 images (100 malign, 100 benign) without data augmentation. We use $BCE_\alpha$ in (4) as the training loss with $\alpha = 0.1$. The models are trained for $5,000$ iterations using AdamW optimizer [27] with learning rate $3 \times 10^{-4}$, weight decay 0.05, and batch size 128. We train five reward models for the ensemble.

---

[11]https://github.com/openai/guided-diffusion
[12]https://pytorch.org/vision/main/models/generated/torchvision.models.resnet18

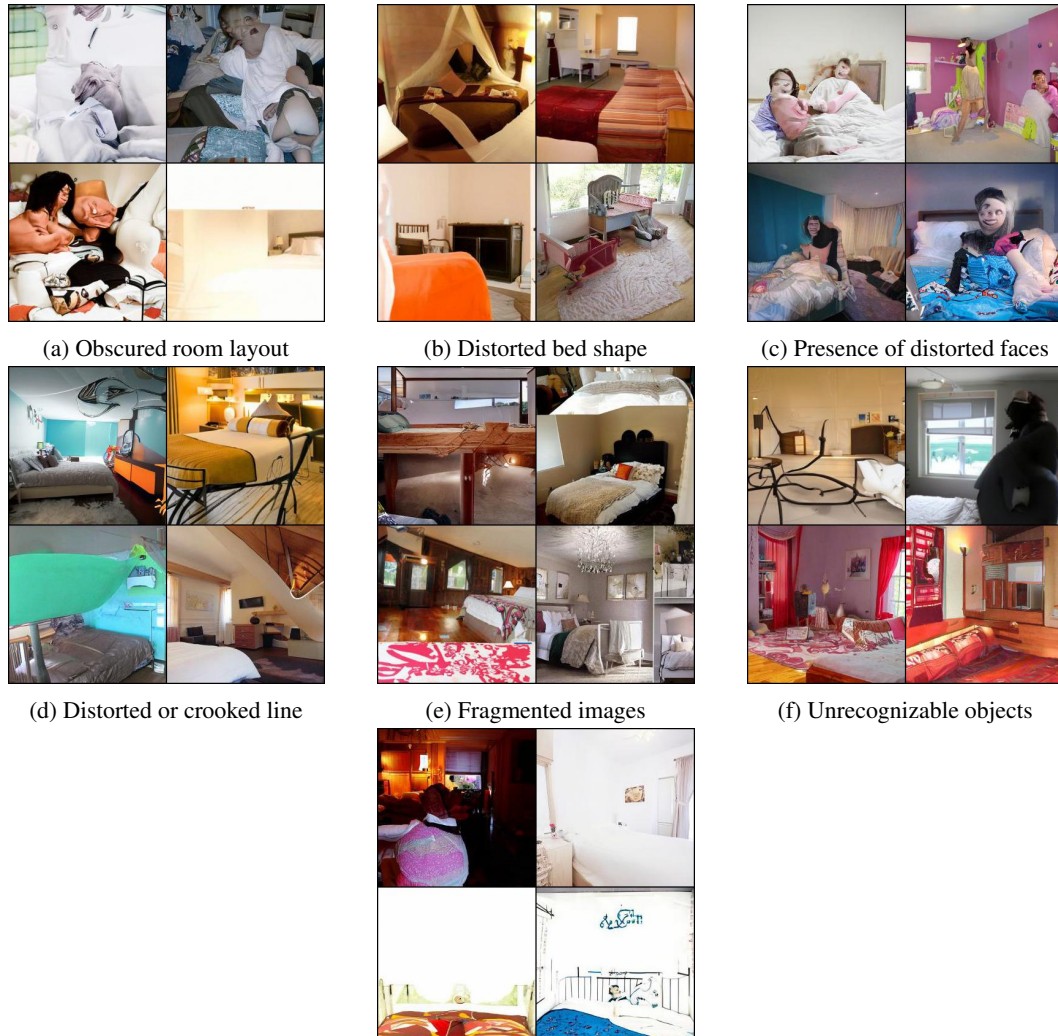

(a) Obscured room layout     (b) Distorted bed shape     (c) Presence of distorted faces

(d) Distorted or crooked line     (e) Fragmented images     (f) Unrecognizable objects

(g) Excessive brightness

Figure 17: Examples of "broken" LSUN bedroom images

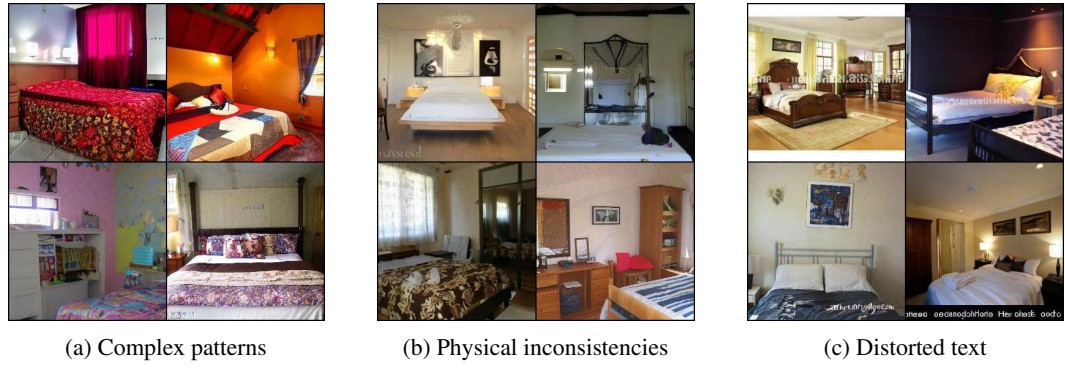

(a) Complex patterns     (b) Physical inconsistencies     (c) Distorted text

Figure 18: Images classified as benign despite giving the impression of being corrupted or damaged.

### K.4 Sampling and ablation study

For sampling via reward ensemble without backward guidance and recurrence, we choose $\omega = 2.0$. We compare the censoring performance of a reward model ensemble with two non-ensemble reward models called "**Single**" and "**Union**" in Figure 4:

- "**Single**" model refers to one of the five reward models for the ensemble method, which is trained on randomly selected 100 malign images, and a set of 100 benign images.
- "**Union**" model refers to a model which is trained on 100 malign images and a collection of $\sim 500$ benign images, combining the set of benign images used to train the ensemble. These models are trained for 15,000 iterations with $\alpha = 0.02$ for the $BCE_\alpha$ loss.

For these non-ensemble models, we use $\omega = 10.0$, which is $K = 5$ times the guidance weight used in the ensemble case. For censored image generation using ensemble combined with backward guidance and recurrence as discussed in Section G, we use $\omega = 2.0$, learning late 0.002, $B = 5$, and $R = 4$.

### K.5 Censored generation samples

Figure 19 shows uncensored, baseline generation. Figures 20–31 present a total of 1,000 images sampled with censored generation, 500 generated by ensemble reward models without backward guidance and recurrence and 500 with backward guidance and recurrence.

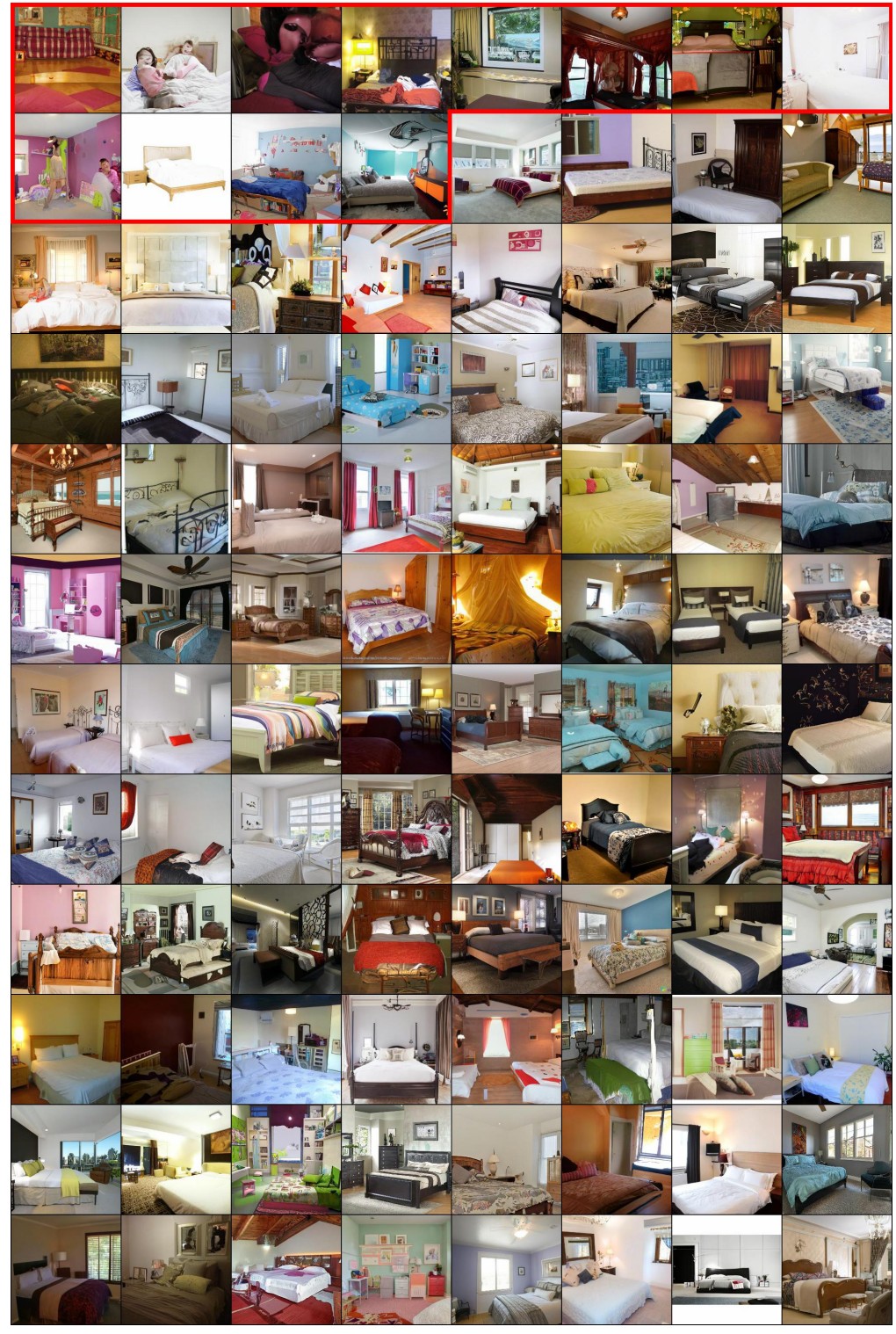

Figure 19: 96 uncensored baseline image samples. Malign images are labeled with red borders and positioned at the beginning for visual clarity.

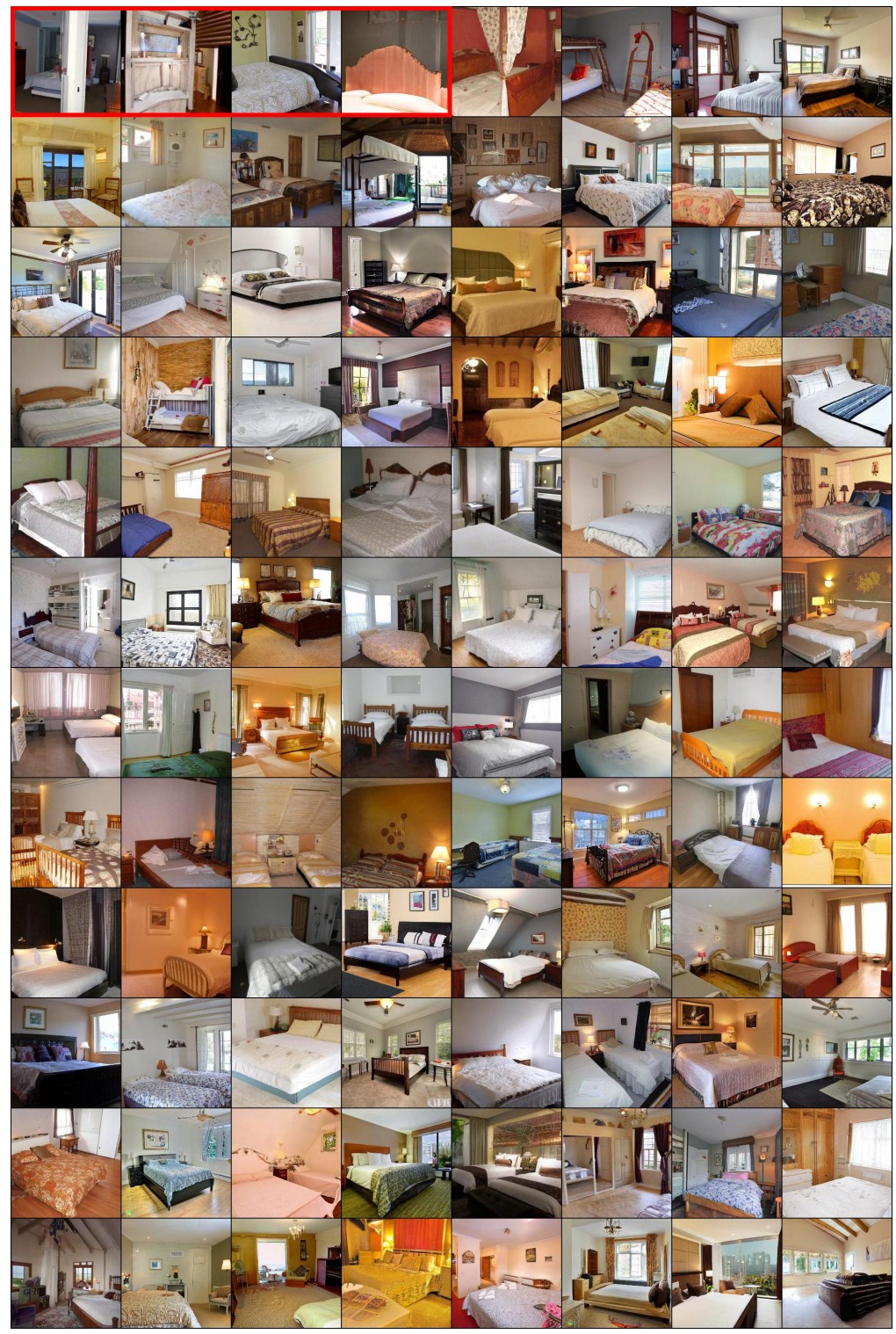

Figure 20: First set (1–96) of images among the 500 non-curated censored generation samples with a reward model ensemble and without backward guidance and recurrence. Malign images are labeled with red borders and positioned at the beginning for visual clarity. Qualitatively and subjectively speaking, we observe that censoring makes the malign images less severely "broken" compared to the malign images of the uncensored generation.

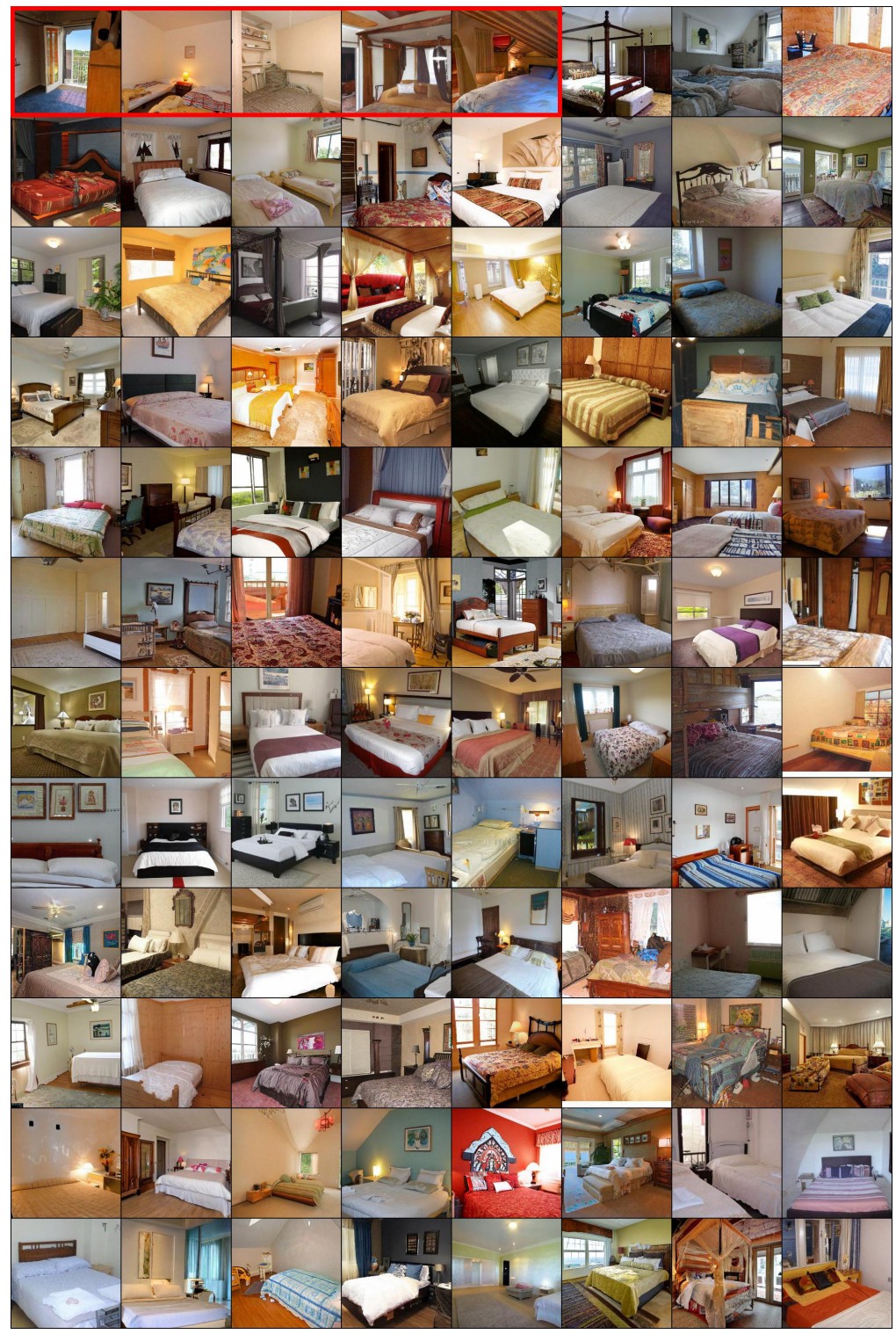

Figure 21: Second set (97–192) of images among the 500 non-curated censored generation samples with a reward model ensemble and without backward guidance and recurrence. Malign images are labeled with red borders and positioned at the beginning for visual clarity. Qualitatively and subjectively speaking, we observe that censoring makes the malign images less severely "broken" compared to the malign images of the uncensored generation.

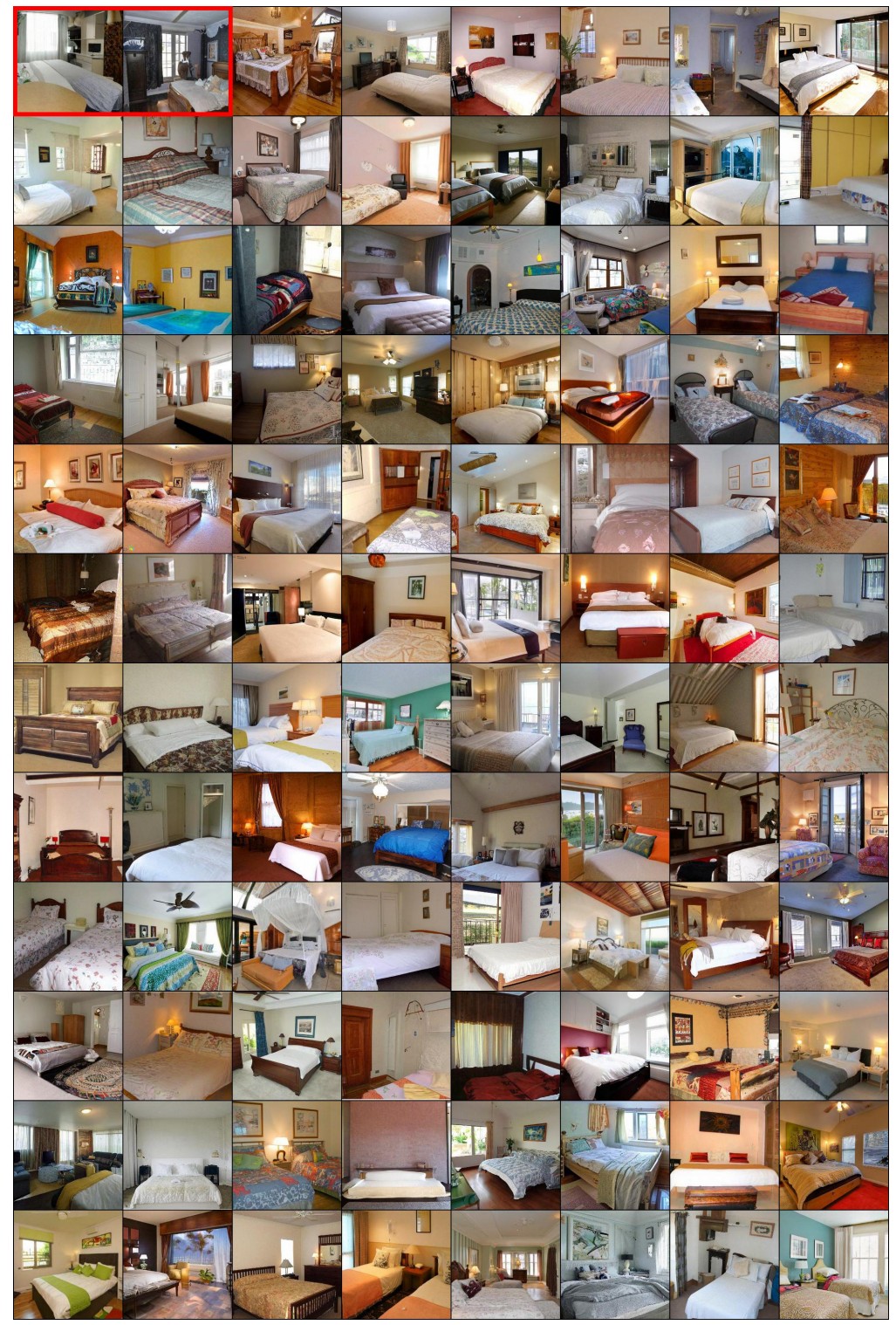

Figure 22: Third set (193–288) of images among the 500 non-curated censored generation samples with a reward model ensemble and without backward guidance and recurrence. Malign images are labeled with red borders and positioned at the beginning for visual clarity. Qualitatively and subjectively speaking, we observe that censoring makes the malign images less severely "broken" compared to the malign images of the uncensored generation.

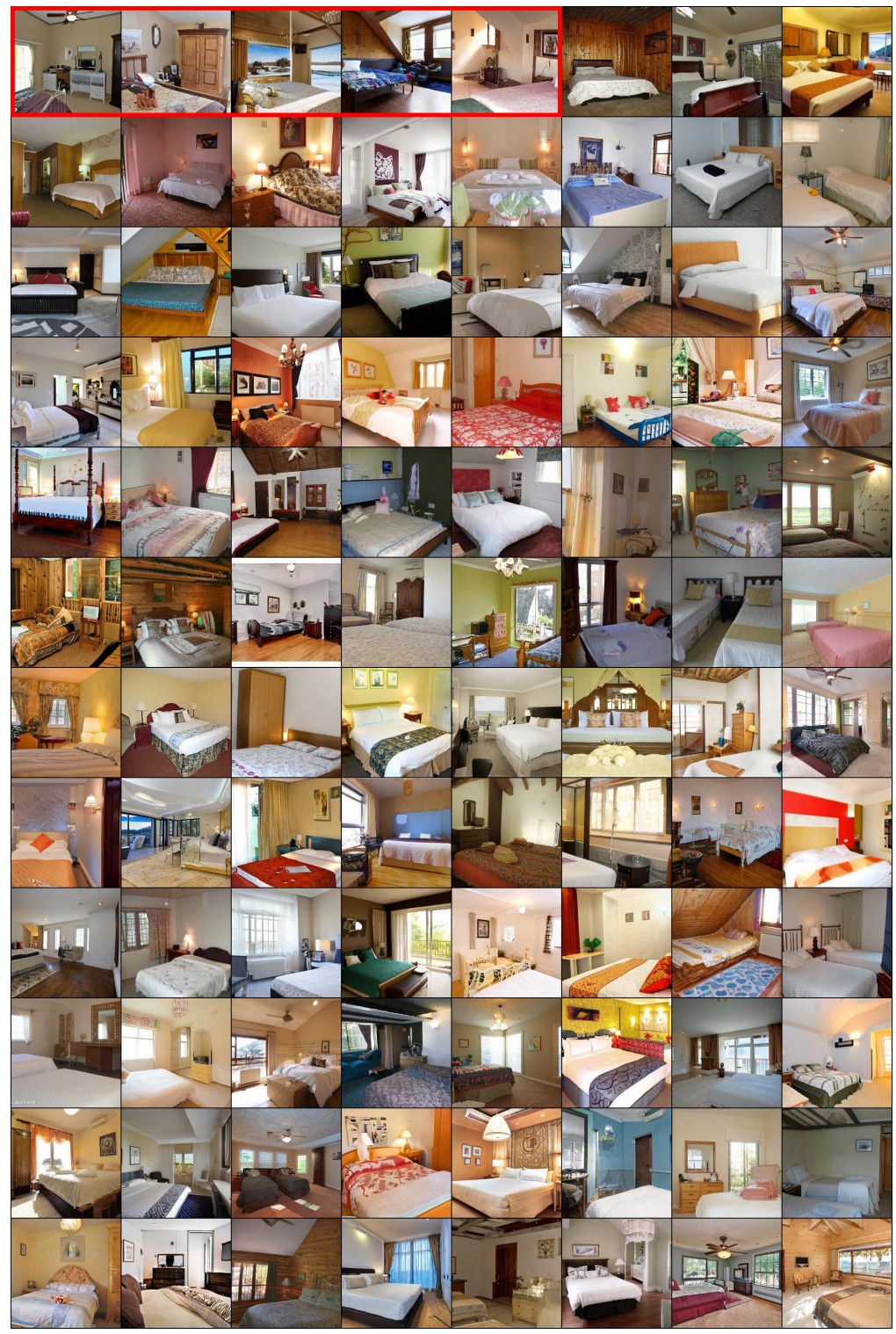

Figure 23: Fourth set (289–384) of images among the 500 non-curated censored generation samples with a reward model ensemble and without backward guidance and recurrence. Malign images are labeled with red borders and positioned at the beginning for visual clarity. Qualitatively and subjectively speaking, we observe that censoring makes the malign images less severely "broken" compared to the malign images of the uncensored generation.

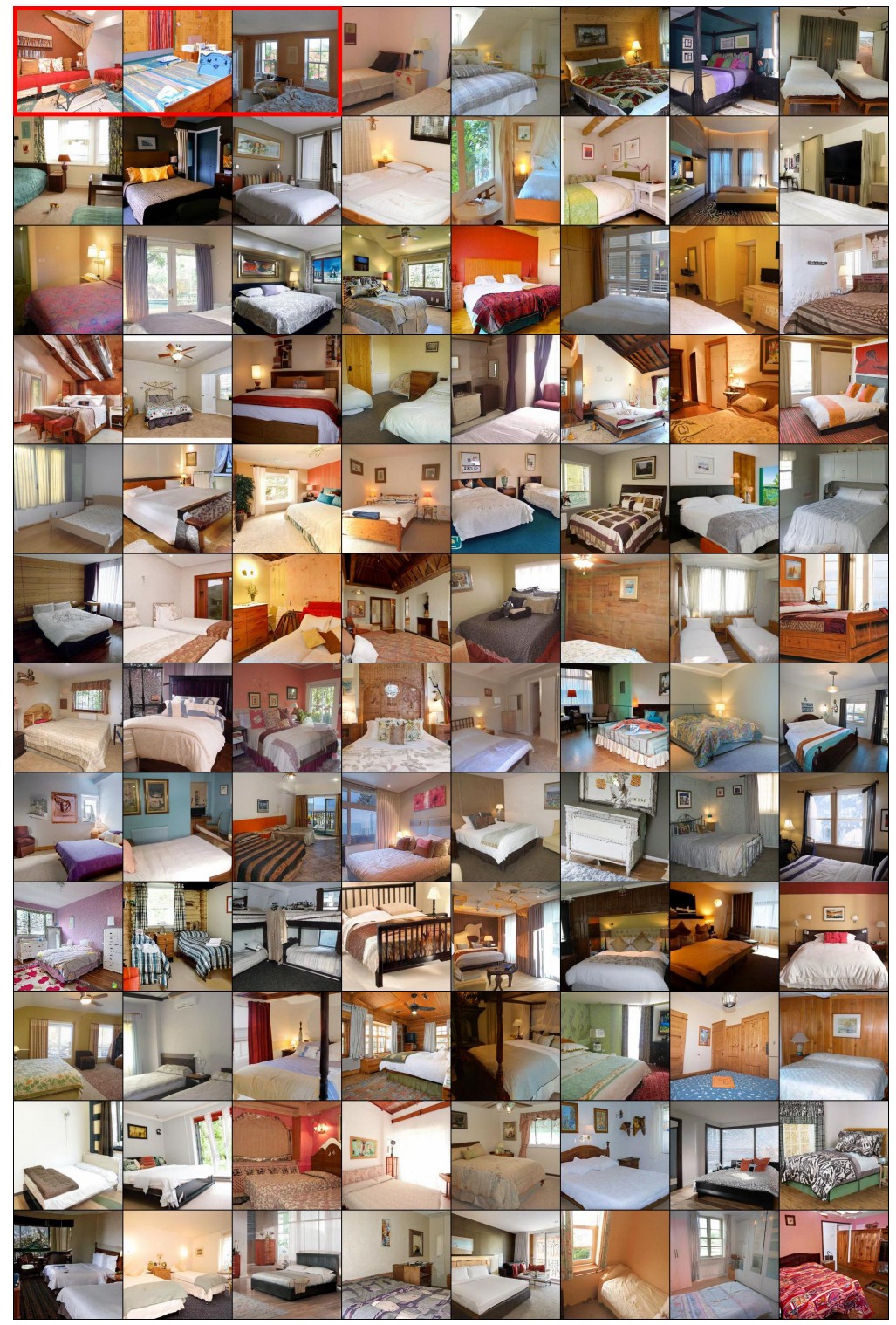

Figure 24: Fifth set (385–480) of images among the 500 non-curated censored generation samples with a reward model ensemble and without backward guidance and recurrence. Malign images are labeled with red borders and positioned at the beginning for visual clarity. Qualitatively and subjectively speaking, we observe that censoring makes the malign images less severely "broken" compared to the malign images of the uncensored generation.

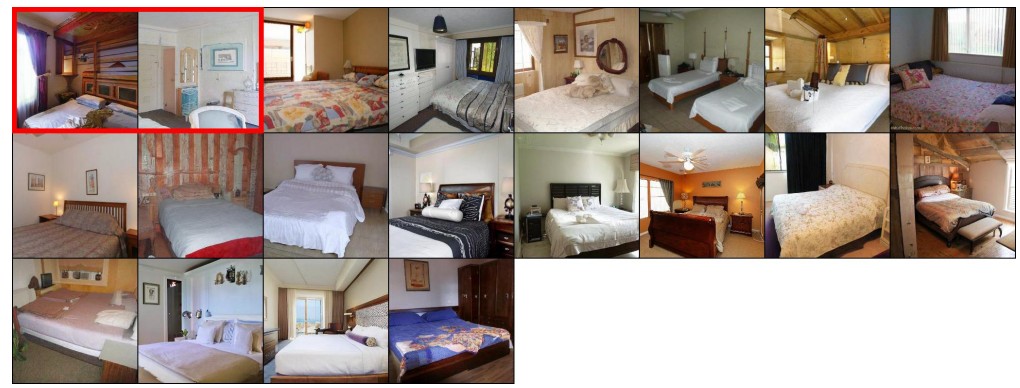

Figure 25: Sixth set (481–500) of images among the 500 non-curated censored generation samples with a reward model ensemble and without backward guidance and recurrence. Malign images are labeled with red borders and positioned at the beginning for visual clarity. Qualitatively and subjectively speaking, we observe that censoring makes the malign images less severely "broken" compared to the malign images of the uncensored generation.

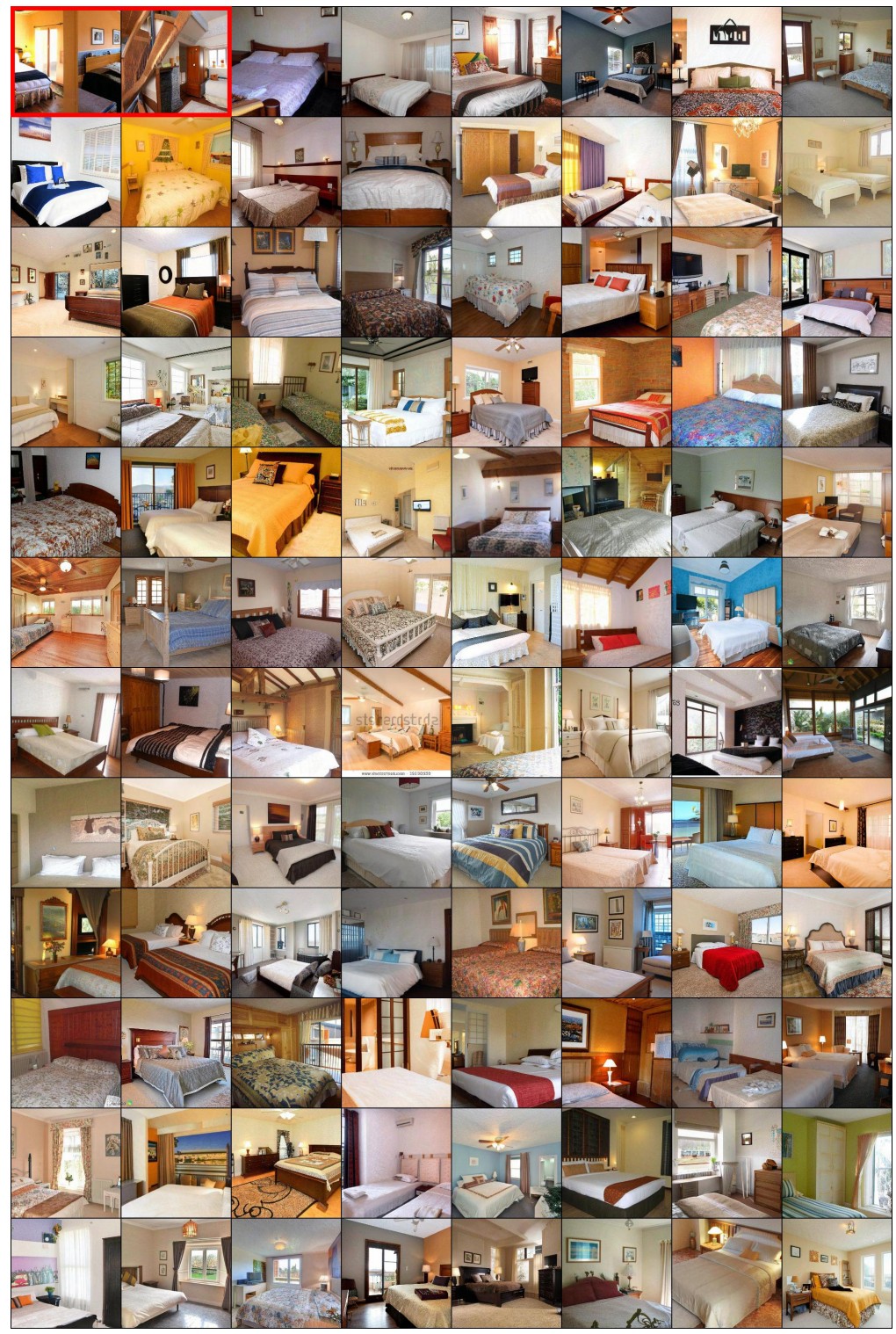

Figure 26: First set (1–96) of images among the 500 non-curated censored generation samples with a reward model ensemble and **with** backward guidance and recurrence. Malign images are labeled with red borders and positioned at the beginning for visual clarity. Qualitatively and subjectively speaking, we observe that censoring makes the malign images less severely "broken" compared to the malign images of the uncensored generation.

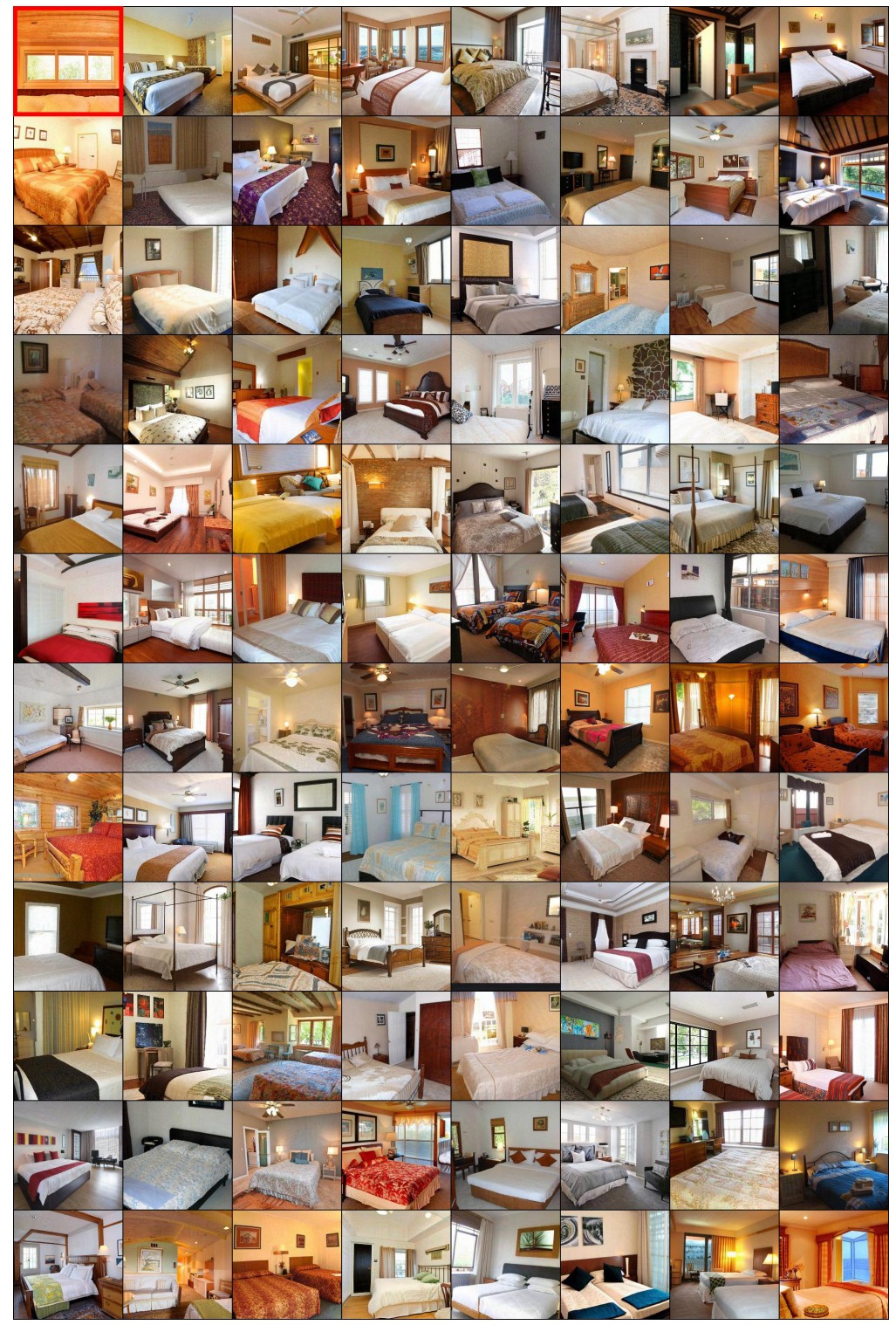

Figure 27: Second set (97–192) of images among the 500 non-curated censored generation samples with a reward model ensemble and **with** backward guidance and recurrence. Malign images are labeled with red borders and positioned at the beginning for visual clarity. Qualitatively and subjectively speaking, we observe that censoring makes the malign images less severely "broken" compared to the malign images of the uncensored generation.

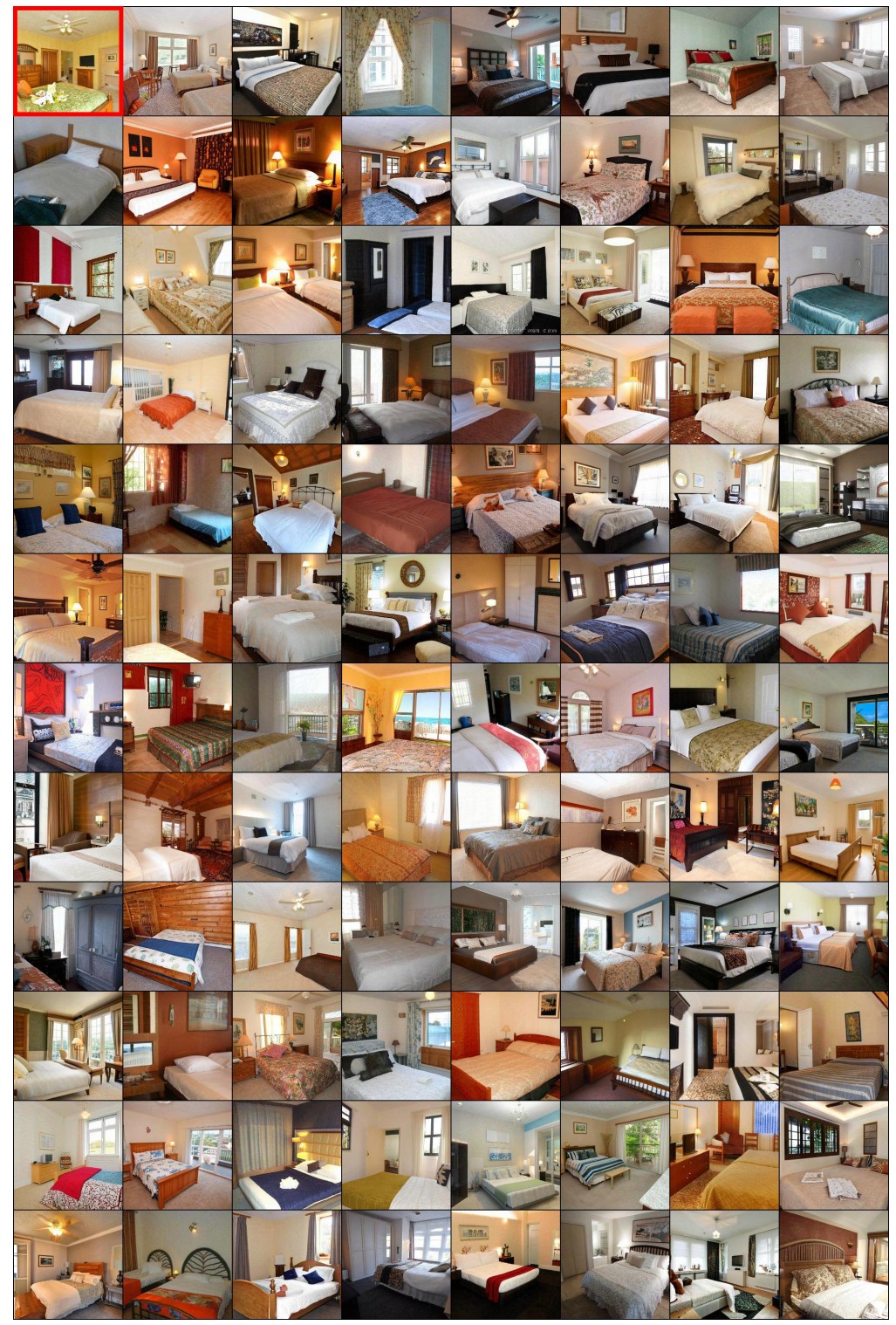

Figure 28: Third set (193–288) of images among the 500 non-curated censored generation samples with a reward model ensemble and **with** backward guidance and recurrence. Malign images are labeled with red borders and positioned at the beginning for visual clarity. Qualitatively and subjectively speaking, we observe that censoring makes the malign images less severely "broken" compared to the malign images of the uncensored generation.

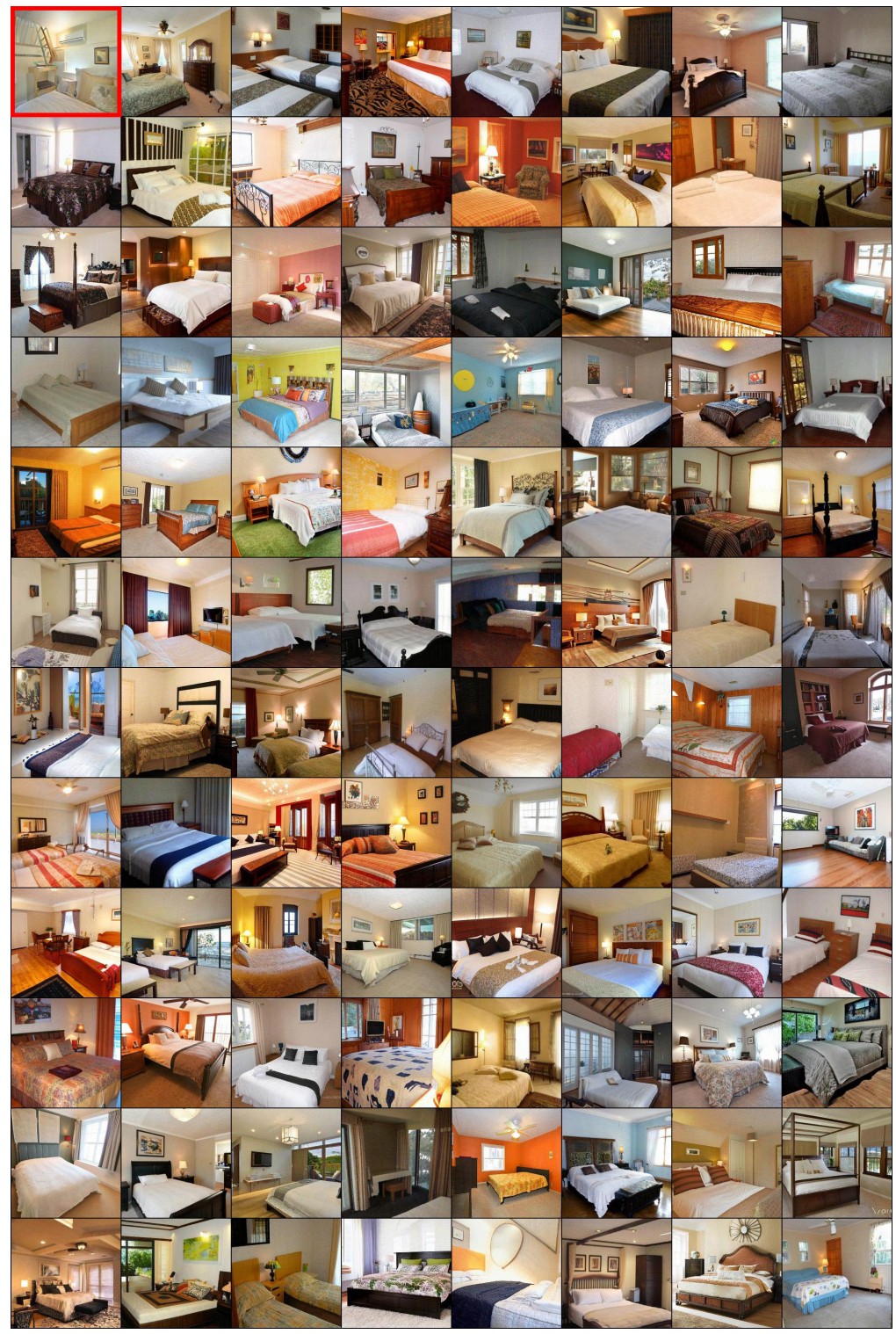

Figure 29: Fourth set (289–384) of images among the 500 non-curated censored generation samples with a reward model ensemble and **with** backward guidance and recurrence. Malign images are labeled with red borders and positioned at the beginning for visual clarity. Qualitatively and subjectively speaking, we observe that censoring makes the malign images less severely "broken" compared to the malign images of the uncensored generation.

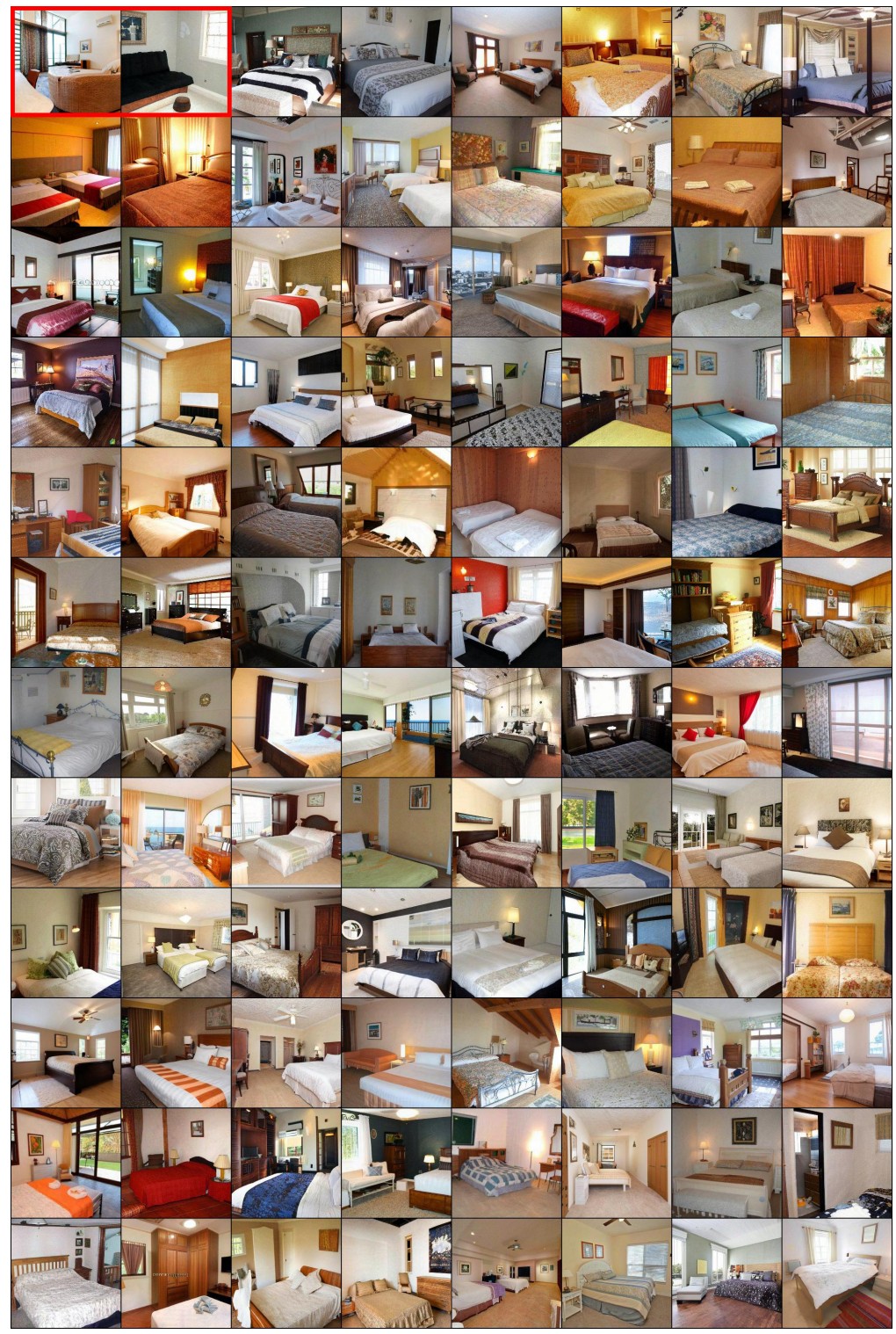

Figure 30: Fifth set (385–480) of images among the 500 non-curated censored generation samples with a reward model ensemble and **with** backward guidance and recurrence. Malign images are labeled with red borders and positioned at the beginning for visual clarity. Qualitatively and subjectively speaking, we observe that censoring makes the malign images less severely "broken" compared to the malign images of the uncensored generation.

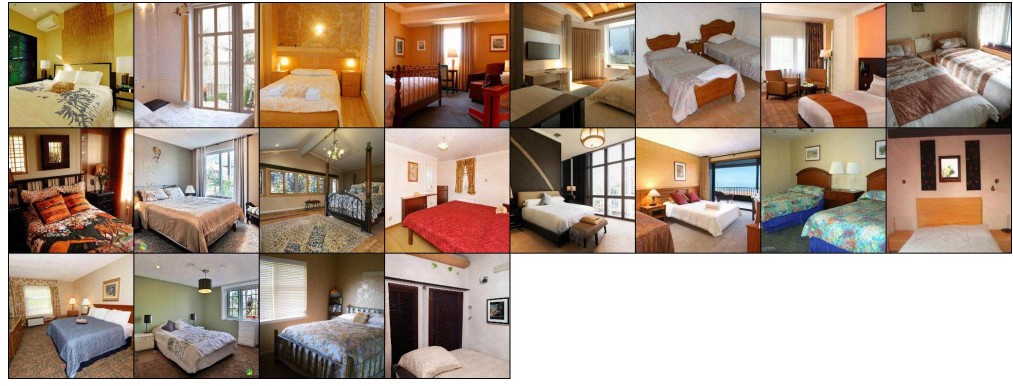

Figure 31: Sixth set (481–500) of images among the 500 non-curated censored generation samples with a reward model ensemble and **with** backward guidance and recurrence. Malign images are labeled with red borders and positioned at the beginning for visual clarity. Qualitatively and subjectively speaking, we observe that censoring makes the malign images less severely "broken" compared to the malign images of the uncensored generation.

# L  Stable Diffusion: Experiment details and image samples

## L.1  Pre-trained diffusion model

We use the pretrained Stable Diffusion[13], version 1.4. We generate images using the default setting, which uses image size of $512 \times 512$ and 50 DDIM [41] steps.

## L.2  Reward model training

We utilize a ResNet18 architecture for the reward model, using the pre-trained weights available in torchvision.models' "DEFAULTS" setting[14], which is pre-trained on the ImageNet1k [10] dataset. We replace the final layer with a randomly initialized fully connected layer with a one-dimensional output. We train all layers of the reward model using the human feedback dataset of 200 images (100 malign, 100 benign) without data augmentation. We use $BCE_\alpha$ in (4) as the training loss with $\alpha = 0.1$. The models are trained for $10,000$ iterations using AdamW optimizer [27] with learning rate $10^{-4}$, weight decay 0.05, and batch size 64.

## L.3  Sampling and ablation study

For sampling via reward ensemble without backward guidance and recurrence, we choose $\omega = 4.0$. We compare the censoring performance of a reward model ensemble with two non-ensemble reward models called "**Single**" and "**Union**" in Figure 5:

- "**Single**" model refers to one of the five reward models for the ensemble method, which is trained on randomly selected 100 malign images, and a set of 100 benign images.
- "**Union**" model refers to a model which is trained on 100 malign images and a collection of $\sim 500$ benign images, combining the set of benign images used to train the ensemble. These models are trained for 30,000 iterations with $\alpha = 0.02$ for the $BCE_\alpha$ loss.

For these non-ensemble models, we use $\omega = 20.0$, which is $K = 5$ times the guidance weight used in the ensemble case. For censored image generation using ensemble combined recurrence as discussed in Section G, we use $\omega = 4.0$ and $R = 4$.

## L.4  Censored generation samples

Figure 32 shows uncensored, baseline generation. Figures 33 and 34 respectively present images sampled with censored generation without and with recurrence.

---

[13]https://github.com/CompVis/stable-diffusion
[14]https://pytorch.org/vision/main/models/generated/torchvision.models.resnet18

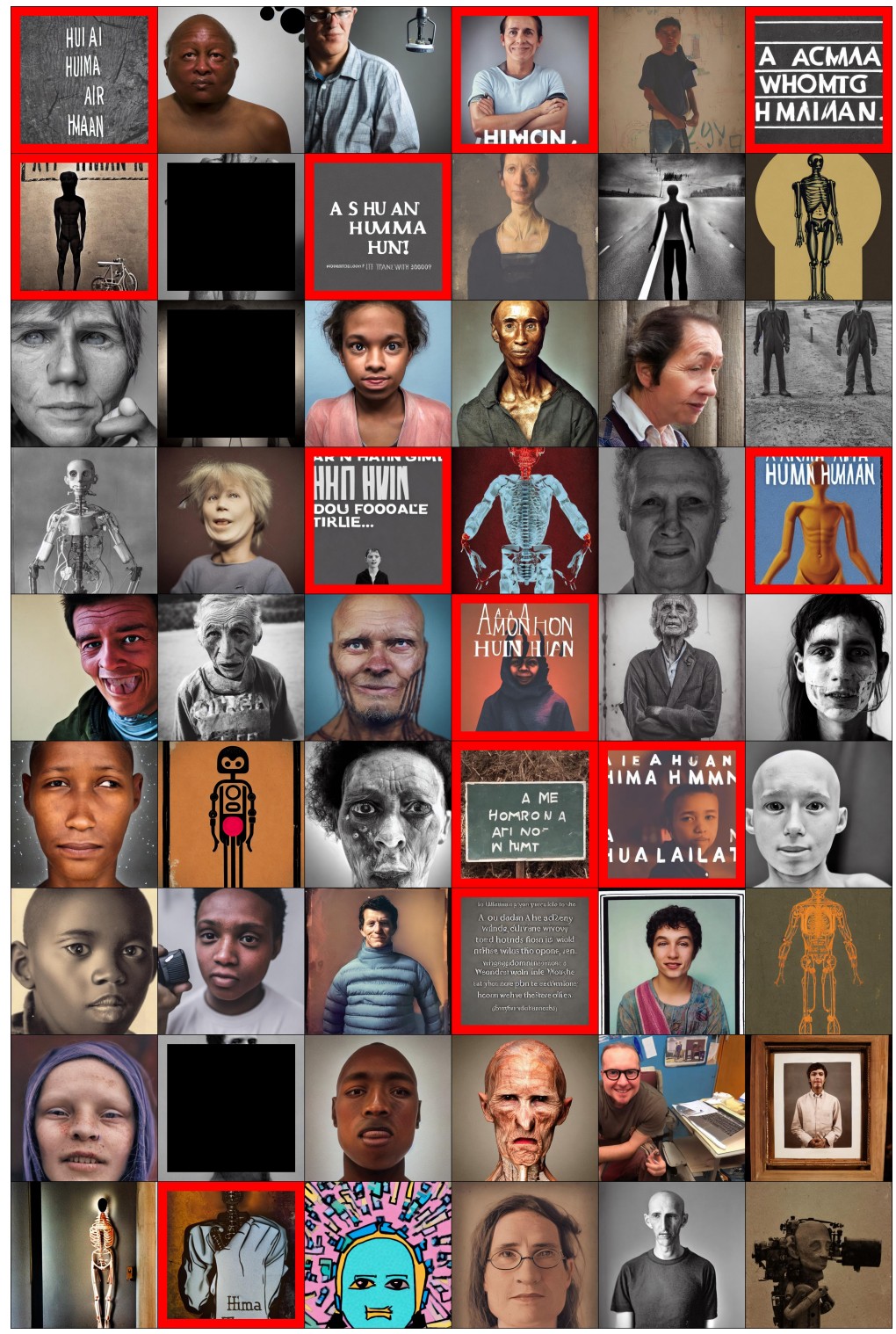

Figure 32: Uncensored baseline image samples. Malign images are labeled with red borders, and images involving undressed bodies are partially blackened out.

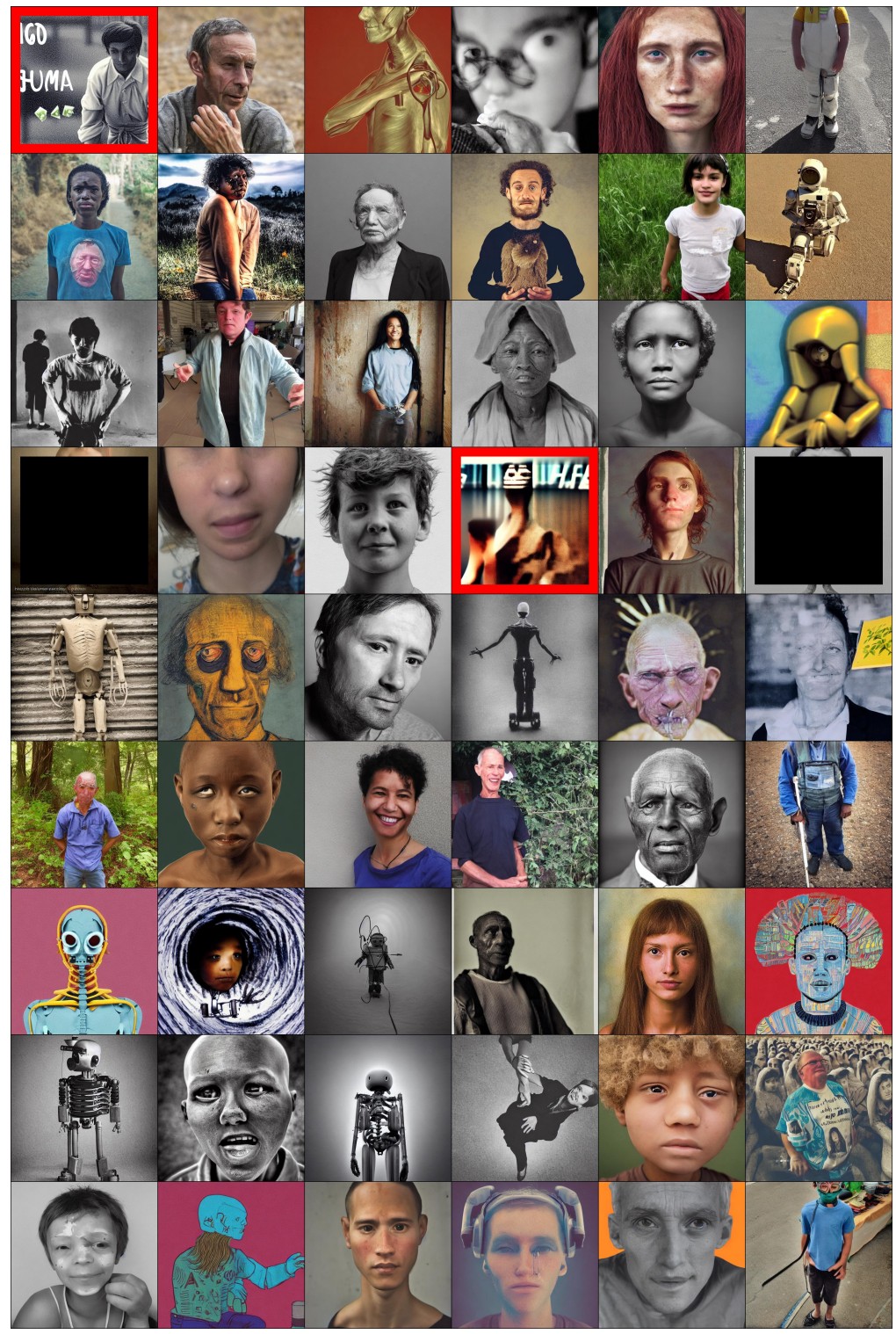

Figure 33: Non-curated censored generation samples without backward guidance and recurrence. Reward model ensemble is trained on 100 malign images. Malign images are labeled with red borders, and images involving undressed bodies are partially blackened out.

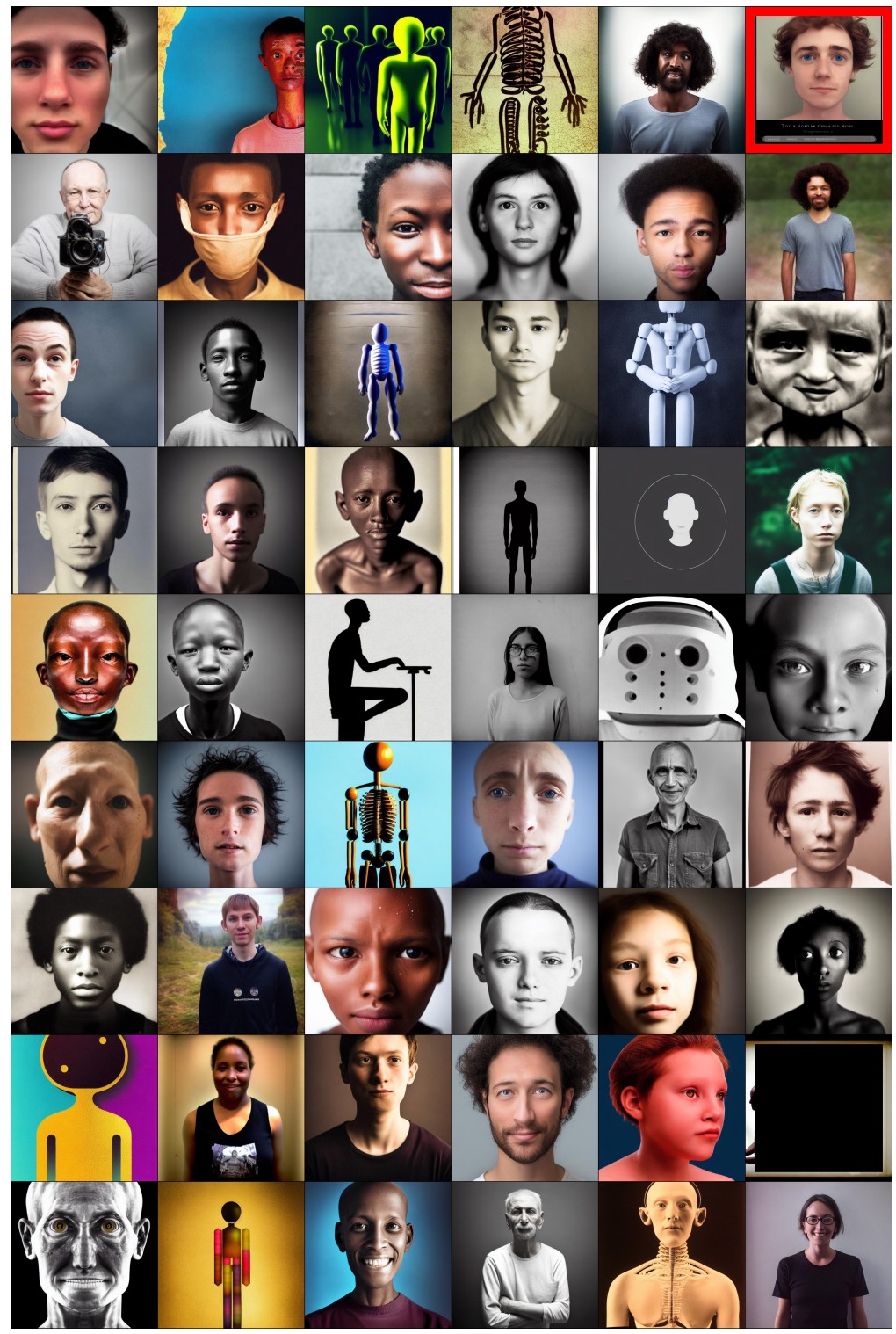

Figure 34: Non-curated censored generation samples **with** recurrence. Reward model ensemble is trained on 100 malign images. Malign images are labeled with red borders, and images involving undressed bodies are partially blackened out.

# M   Transfer learning ablation

To demonstrate the necessity of transfer learning for relatively more complex tasks, we compare it with training reward model from scratch. We consider the LSUN bedroom task of Section 5.4. We randomly initialize the weights of time-dependent reward model with half-UNet architecture and train it for 40,000 iterations with batch size 128. We use the training loss $BCE_\alpha$ with $\alpha = 0.1$ and use the guidance weight of $\omega = 10.0$ for sampling.

We observe that censoring fails without transfer learning, despite our best efforts to tune the parameters. The reward model successfully interpolates the training data, but its classification performance on the test dataset (which we create separately using additional human feedback data) is poor: it shows 70.63% and 43.23% accuracy respectively for malign and benign test images. If we nevertheless proceed to perform censored generation using this reward model (trained without transfer learning), the proportion of malign images is $15.68\% \pm 5.25\%$ (measured using 500 generated images across 5 independent trials). This is no better than 12.6% of the baseline model without censoring.

# N    Using malign images from secondary source

In this section, we demonstrate the effectiveness of our framework even with malign images from a secondary source, instead of model-generated images. This strategy may be useful, e.g., in cases where the baseline model rarely generates malign images (making it difficult to collect a sufficient number of them) but a user still wishes to impose hard restriction of not generating certain images.

We consider the setup of Section 5.2, and repeat the similar procedure except that we use malign images manually crawled from the Shutterstock webpage[15]; we utilize Microsoft Windows Snipping Tool to take screenshots of "church", "gothic church" and "cathedral" images with clearly visible watermarks, resize them to $256 \times 256$ resolution and apply random horizontal flip. We use a fixed set of these 30 secondary malign images (Figure 35) and 30 benign images generated by the baseline model to train each reward model. We use the same hyperparameters as in Section I for reward training and sampling, except the only difference of using the guidance weight $\omega = 1.0$.

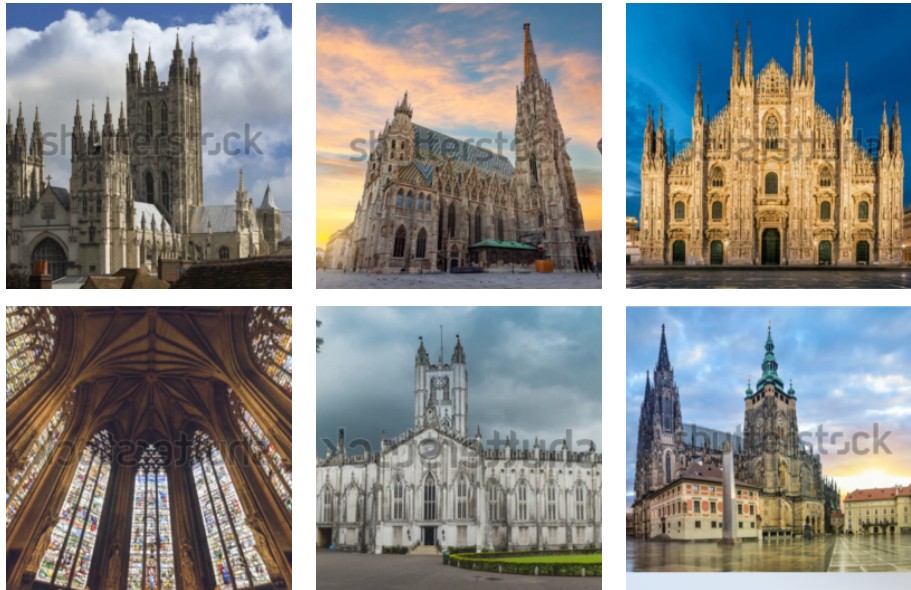

Figure 35: Examples of malign (watermarked) church images, collected manually from a secondary source other than the baseline model.

|  | Baseline (No censoring) | Ordinary censoring | Secondary censoring |
|---|---|---|---|
| Malign proportion | 11.4% | 0.76% | 1.4% |

Table 1: Performance comparison between ordinary censoring vs. secondary censoring (using secondary malign images in place of model-generated malign images). Both censoring setups use ensemble of 5 reward models and recurrence with $R = 4$.

We display the result of censoring using secondary malign images in Table 1 (labeled "secondary censoring"). We ensemble 5 reward models (trained on the same set of 30 secondary malign images) and apply recurrence with $R = 4$. Recall that the baseline model generates 11.4% malign images, while with secondary censoring, the proportion drops to 1.4%. This is slightly worse than 0.76% of our ordinary censoring methodology (Figure 2), but still shows that secondary malign images could be effective alternatives to model-generated images in certain setups.

---

[15]https://www.shutterstock.com/

