# OpenReview forum: "Censored Sampling of Diffusion Models Using 3 Minutes of Human Feedback"
_NeurIPS.cc/2023/Conference — NeurIPS 2023 poster_

### Official Review · Reviewer_6Ndg · 2023-07-02

**Soundness:** 3 good
**Presentation:** 4 excellent
**Contribution:** 3 good
**Rating:** 6
**Confidence:** 4

**Summary:**

The work aims to solve censored diffusion sampling problem, which prevent diffusion model generating malign / bad images. The core approach is to train a classifier and apply classifier-guided diffusion generation.

**Strengths:**

1. Authors present an interesting finding that the classifier-guided diffusion can effectively reduce the probability of generating malign images.
2. Authors present two approaches to learn the classifier, ensemble-based approaches when number of malign images less than benign images, imitation learning for when malign images are dominates.
3. The proposed methods are lightweight and can be effective with small labeled datasets.

**Weaknesses:**

The core technique employed in the work is not new.  And compared with existing works that are based on text2image diffusion models, authors use relatively simple diffusion models that are either class-condition diffusion models or diffusion models for one category dataset. I am not sure the effectiveness and efficiency of the proposed method can be scaled to a large diffusion model that handles for complicate diffusion models. The classifiers in three experiments are relatively easy and that may explain why it is so effective. It may not be true in large text2image settings.

**Questions:**

It would be more convincing that the authors provide more experiments for more complex diffusion models, such as stable diffusion.

**Limitations:**

See above

---

> ### Author Rebuttal · Authors · 2023-08-09
>
> We thank the reviewer for constructive comments. As suggested by the reviewer, we added an experiment using the **Stable Diffusion** model. Please refer to Section 2 of the common rebuttal and the attached pdf document for details.
> We plan to add this new experiment, with some polishing, to the later version of the paper.
> For now, we hope that the reviewer's concern on the applicability of our framework to larger/complex models is addressed, and if that is the case, we kindly ask the reviewer to consider increasing the rating.
>
> In terms of the technical novelty, we acknowledge that the building blocks of our method are not new, but we ask the reviewer to consider that the main contribution of our paper is to **provide a simple, general, and easily adaptable framework that exhibits extreme sample/feedback efficiency.**
> A priori, it was not at all obvious that this level of sample efficiency is possible.
> Considering that both diffusion models and RLHF are currently actively investigated topics within the literature, we do believe that these findings will be useful to practitioners.

---

> > ### Comment · Reviewer_6Ndg · 2023-08-16
> >
> > Thanks for updating. The new experiments look promising. I raise my score.

---

### Official Review · Reviewer_GFro · 2023-07-03

**Soundness:** 4 excellent
**Presentation:** 4 excellent
**Contribution:** 4 excellent
**Rating:** 6
**Confidence:** 3

**Summary:**

The authors examine the problem of preventing the generation of certain types of images generated by a diffusion model. To achieve this, the authors propose using a reward model trained on human feedback. The authors demonstrate their approach from examples that require minimal human feedback to achieve sufficient censoring performance.

**Strengths:**

- The authors make a good argument in that 3 minutes of human feedback can be more cost-efficient than retraining models.
- Their discussion and experiments in benign-dominant and malign-dominant scenarios are thorough.
- Their ablation studies in the experiments give convincing arguments that human feedback works well.

**Weaknesses:**

- A discussion of when to use time-dependent or time-independent would have been nice.
- In the LSUN bedroom experiment, it took 15 minutes so it doesn't always take 3 minutes.

**Questions:**

- Line 52: there are two consecutive "the".
- Figure 3(b) text: "Comparsion" to "Comparison"
- How far can this method go? I'm sure there are practioners that want 0% malign images.

**Limitations:**

- The authors do not explicitly state their limitations.

---

> ### Author Rebuttal · Authors · 2023-08-10
>
> We highly appreciate the constructive comments and the positive evaluation from the reviewer. We made our best efforts to reflect the comments to improve the paper. In the following, we address each of the reviewer's concerns in detail.
>
> ### 1. On time-dependent vs. time-independent guidance
> Guidance using time-dependent reward $r(\cdot, t)$ is conceptually simpler (same as the usual classifier guidance) but requires a time-dependent architecture and the reward model should be trained on images with different level of signal-to-noise ratios.
> For these reasons, one usually has to train the model from scratch, which is more suitable for simpler/smaller tasks (Sections 5.1 and 5.3).
> On the other hand, time-independent reward models require additional techniques such as universal guidance, but the reward model has to be trained only on clean images, so one can exploit various pre-trained classifiers and expedite the reward training through transfer learning.
> This makes (time-independent) transfer learning more suitable for complex/large-scale tasks (Sections 5.2, 5.4 and newly added Stable Diffusion experiment).
> We found in our preliminary experiments that for these tasks, it was not easy to train time-dependent reward models for successful censoring, while transfer learning converged faster and provided promising results.
> We will add discussion on this point in a later version of the paper.
>
> ### 2. Regarding the human time for the LSUN bedroom task
> Please refer to the discussion on this point in Section 3 of the common response.
> We further emphasize that our techniques are applicable to solving a task on the **Stable Diffusion** model, (also shown in the common response), and this actually **requires only 3~4 minutes of human feedback, despite the task's complexity.**
>
> ### 3. On how far the iterative application of our method could reach
> We expect that iterative application of our methods (Algorithm 2 in particular) with progressively more feedback data **will achieve 0% malign images in the limit.**
> Algorithm 2 iterates the process of "collecting additional human feedback data $\rightarrow$ reward training $\rightarrow$ (censored) sampling using the reward", and the 3 rounds, without much feedback data, can already reduce the initial malign proportion dramatically (from 68.6% to 2.2% and 1.0%, without and with universal guidance components, in Section 5.3).
> During the rebuttal period, we tried **another round (Round 4) of imitation learning and observed another significant reduction of malign proportion to 1.38% and 0.58%, without and with universal guidance.**
> Fitting the tendency, it seems that further rounds will steadily reduce the probability of malign generation and eventually converge to zero (Figure 3a in the attached pdf), but this will come at the cost of requiring more compute for data generation and more human feedback due to the high precision of censoring.
> Nevertheless, we do see hope in achieving an extreme level of precision at the expense of additional human feedback and computation.

---

> > ### Comment · Reviewer_GFro · 2023-08-16
> >
> > Thank you very much for your response. You have definitely cleared up my questions.

---

### Official Review · Reviewer_iXGj · 2023-07-08

**Soundness:** 3 good
**Presentation:** 3 good
**Contribution:** 2 fair
**Rating:** 6
**Confidence:** 4

**Summary:**

This paper presents censored diffusion model training by using a reward model trained using human feedback. Towards this, the paper utilizes reward model ensembles (for benign dominant settings) and tools from imitation learning (for malign dominant settings).

**Strengths:**

The human feedback part is pretty interesting and is a topic of increasing interest. The empirical results appear to be pretty compelling, though, I am not an expert with respect to evaluating this in particular, so I cannot comment on this with high confidence.

==> post rebuttal: I increased the score from a 5 to a 6.

**Weaknesses:**

The techniques utilized by the paper are pretty well known and in that sense, this paper's contributions appear incremental.

One other question that appears to not be addressed is - by performing this paper's procedure iteratively (as multiple rounds), would the probability of producing malign images be brought down to zero? What are the other ways of achieving this?

Another weakness of the paper is that it is not clear to me if the proposed approach has been compared against other concept removal methods in the literature (which I agree I am not an expert on), but, this is something the authors must either present clarifications on, or, provide more comparisons on.

**Questions:**

a. i wonder if the malign dominant settings can be handled by flipping the reward to be something resembling 1-\prod_{i=1}^K (1-r_{\psi_k}^{k})?

b. does it help to train every single reward model with some form of hard negative mining? I mean this in the context of algorithm 1.

**Limitations:**

I believe the authors must present a discussion surrounding this in the context of their work - what it seeks to address, what remains to be addressed, and how much to trust the proposed algorithm as to whether it truly achieves what it sets out to. I am not certain about going through an ethics review, and I hope the AC and the authors try to see what is necessary in this direction as it clearly seems like there is potential for fleshing some of these concerns out.

---

> ### Author Rebuttal · Authors · 2023-08-10
>
> We appreciate the constructive comments and the positive evaluation from the reviewer.
> In the following, we make our best efforts to address each of the concerns and questions.
> For the concern regarding the comparison against other methods, please refer to Section 4 within our common response.
>
> ### 1. Regarding the contribution of the paper
> In terms of the technical novelty, we acknowledge that the building blocks of our method are not new, but we ask the reviewer to consider that the main contribution of our paper is to **provide a simple, general, and easily adaptable framework that exhibits extreme sample/feedback efficiency.** A priori, it was not at all obvious that this level of sample efficiency is possible. Considering that both diffusion models and RLHF are currently actively investigated topics within the literature, we do believe that these findings will be useful to practitioners.
>
> ### 2. On how far the iterative application of our method could reach
> We expect that iterative application of our methods (Algorithm 2 in particular) with progressively more feedback data **will achieve 0\% malign images in the limit.**
> Algorithm 2 iterates the process of "collecting additional human feedback data $\rightarrow$ reward training $\rightarrow$ (censored) sampling using the reward", and the 3 rounds, without much feedback data, can already reduce the initial malign proportion dramatically (from 68.6% to 2.2% and 1.0%, without and with universal guidance components, in Section 5.3).
> During the rebuttal period, we tried **another round (Round 4) of imitation learning and observed another significant reduction of malign proportion to 1.38% and 0.58%, without and with universal guidance.**
> Fitting the tendency, it seems that further rounds will steadily reduce the probability of malign generation and eventually converge to zero (Figure 3a in the attached pdf), but this will come at the cost of requiring more compute for data generation and more human feedback due to the high precision of censoring.
> Nevertheless, we do see hope in achieving an extreme level of precision at the expense of additional human feedback and computation.
>
> ### 3. On the question of handling malign-dominant tasks
> The reviewer's question is whether the reward ensembling (Algorithm~1) can be used for malign-dominant tasks by switching the roles of malign and benign images (that is, fixing a set $\mathcal{B}$ of benign images and then ensembling $K$ reward models trained on $\mathcal{B}$ together with differently subsampled malign images).
> In principle, one can train reward models $r_{\psi_k}^{(k)}$ in this way and use the ensembled product $r_\psi = \prod_{k=1}^K r_{\psi_k}^{(k)}$ as usual; there is no need to flip the reward because we still expect the product to encourage unanimous approval.
> However, we observe that this is not as effective as imitation learning when the same number of total human feedback data are used; Round 3 of imitation learning that uses total 60 human feedbacks achieves **2.2%** censoring precision without universal guidance, while ensembling 5 reward models trained using 10 fixed benign samples and different sets of 10 malign samples (which also uses 60 feedbacks in total) achieves **6.6%** (please also refer to the discussion related to this point in lines 135--138 on page 6 of the paper).
>
> ### 4. Regarding the use of hard negative mining
> As far as we understand, hard negative mining in this context would mean training the reward model using their false positives, which are the malign generated images even with the censoring technique applied using that reward model. We believe that this process is essentially our Algorithm 2 (imitation learning), and we have already **verified through Experiments in Section 5.3 on Imagenet Tench images that imitation learning is indeed effective, providing improved censoring precision compared to non-imitation learning using the same number of total samples (not using hard negative mining).** Therefore, we expect refining each individual reward model using Algorithm 2 and then applying Algorithm 1 (ensembling) will certainly result in better censoring performance. We, however, do not pursue this direction in depth to focus on delivering of the core ideas.

---

> > ### Comment · Reviewer_iXGj · 2023-08-18
> > **Re. author response**
> >
> > Thanks to the authors for their clarifications. My only remaining concern is whether is it possible to make a (formal) claim about a certified ability of the learnt model to prevent generating mis-aligned images. It will be worthwhile hearing the author's perspective on this, or include a discussion surrounding this in the paper. That said, I will increase my score, and thank the authors for their clarifications.

---

> > > ### Author Response · Authors · 2023-08-21
> > >
> > > We highly appreciate the reviewer's positive evaluation on our results and responses.
> > >
> > > We assume that there exists a ground-truth function $r(x)$ that defines the likelihood of an image $x$ being benign. In principle, with infinite data and an infinitely expressive neural network, we should be able to learn $r(x)$. However, the data we have access to is highly limited in practice, so our goal is to achieve high censoring precision rather than to have any formally certified guarantees.
> > > However, the idea of using the set of techniques from formal verification certifying certain input-output relationships of neural networks [1] seems intriguing.
> > > We are not aware of analogous verification results regarding generative models, but this is certainly a very interesting direction of future work.
> > >
> > > [1] Liu et al., Algorithms for Verifying Deep Neural Networks. Foundations and Trends in Optimization, 2021.

---

### Official Review · Reviewer_zmCo · 2023-07-09

**Soundness:** 2 fair
**Presentation:** 3 good
**Contribution:** 3 good
**Rating:** 6
**Confidence:** 3

**Summary:**

This paper studies the problem of preventing the generation of unwanted images in diffusion models. It formulates the task of 'censoring' and proposes using reward model trained from human labelling to guide the diffusion model. The method requires no fine-tuning and a few minutes of human feedback, while displaying significant reduction of unwanted images in four experiment setups.

**Strengths:**

- This work identifies and formulates an important problem of misalignment in diffusion probabilistic models, which underwent few studies in the past. Solution to the problem can significant improve the utility of generative models and lead to positive social impact.

- RLHF is an emerging technology which is theoretically sound and empirically useful. The proposed method successfully bring the idea of RLHF into diffusion models while combining the idea of classifier guidance. The method itself is clean and simple to train, and requires few modifications to the sampling process.

- From the experiments, the method seem also quite efficient: (i) it sample efficient: requiring up to 100 malign samples, (ii)  it does not need fine-tuning; (iii) it works under limited human feedback, taking only a few minutes of human labeling. These properties are desirable in practical implementation.

**Weaknesses:**

- Under the general framework of training reward models and sampling with guidance, the authors use four different methods to train the reward model for the four different tasks in the paper. The methods are quite heuristic. It appears that for any new concept and dataset, it still requires a manual selection of the training algorithms for the reward models to achieve the best effect, which can be costly. However, the paper does not provide a conclusive algorithm or any unifying principles for the selection. In this sense the study seems unfinished.

- There are no large-scale, principled comparisons with baselines. Even if there are limited existing works studying the problem, I think there are still some naive methods such as using natural language prompts, fine-tuning, and post-selection with classifier. It is not straightforward to see why the method proposed in this paper is the best.

**Questions:**

- Why is it better to use time-independent reward model for censoring watermarks and time-dependent reward model for other tasks?  Is there a general reason why time-independent reward models work better for some tasks and not others?

- Any guidance of choosing $\omega$ ?

- It appears from the paper that the training of reward models need large datasets containing images related to a single unwanted concept (for example fish with human faces). If I do not have such high-quality, single-purpose datasets, maybe only random images with all kinds of undesirable concepts, can the method still work well?

**Limitations:**

See above.

---

> ### Author Rebuttal · Authors · 2023-08-10
>
> We highly appreciate the constructive comments and the positive evaluation from the reviewer.
> We made our best efforts to reflect the comments to improve the paper.
> In the following, we address each of the reviewer's comments in detail.
>
> ### 1. Regarding the principles for selecting the techniques
> We believe that our strategy on when to apply which techniques can be summarized into the following **two simple "rules"**:
> - **Ensemble vs. Imitation learning.** We classify the censoring tasks into benign-dominant and malign-dominant cases, and apply Algorithm 1 (reward ensembling) to the former and Algorithm 2 (imitation learning) to the latter. Please also refer to our discussion on this point in lines 132--138 on page 6 of the paper.
> - **Time-dependent vs. time-independent reward models.** Guidance using time-dependent reward $r(\cdot, t)$ is conceptually simpler (same as the usual classifier guidance) but requires a time-dependent architecture and the reward model should be trained on images with different level of signal-to-noise ratios. For these reasons, one usually has to train the model from scratch, which is more suitable for simpler/smaller tasks (Sections 5.1 and 5.3). On the other hand, time-independent reward models require additional techniques such as universal guidance, but the reward model has to be trained only on clean images, so one can exploit various pre-trained classifiers and expedite the reward training through transfer learning. This makes (time-independent) transfer learning more suitable for complex/large-scale tasks (Sections~5.2, 5.4 and newly added Stable Diffusion experiment). We found in our preliminary experiments that for these tasks, it was not easy to train time-dependent reward models for successful censoring, while transfer learning converged faster and provided promising results.
>
> Besides these rules, we also utilize the universal guidance components to boost the censoring performance, but they apply globally to all tasks that we consider.
> Additionally, we observe that our framework requires minimal hyperparameter tuning; the typical values $\alpha=0.1$ (parameter used in the reward-training loss $BCE_\alpha$), reward learning rate $3\times 10^{-4}$, batch size 128, and number of recurrence $R=4$ work sufficiently well for most setups.
> For the choice of guidance weight $\omega$, we provide a separate discussion below.
> We hope that our clarification resolves the reviewer's impression that there is no principled strategy on the choice of methods.
>
> ### 2. Regarding comparison with other methods
> Please refer to our common response (Section 4 therein) on this point.
>
> We provide comparisons with rejection sampling (post-selection in the reviewer's terminology) using the reward model in Sections~5.3 and 5.4, where it is shown that rejection sampling is not only worse than the guidance method in terms of the precision, but also, it rejects a large proportion of generated samples, which is undesirable.
>
> We use non-text conditional models in the paper's experiments so direct comparison to prompting was not available, but in our additional experiment using the Stable Diffusion model, some basic attempts like adding "without text" or "no text" to the prompts did not produce successful results.
> We acknowledge, however, that sophisticated prompt engineering, negative prompting, or adding specific requests regarding the background could also prevent the appearance of texts.
> At this stage, we view our experiment on Stable Diffusion as a proof of concept that our simple methodology does work on large-scale text-to-image diffusion models (rather than claiming superiority over all possible alternatives).
> Carefully identifying the problems where even dedicated trials using natural language prompts fail, and solving them through human feedback, seems to be an interesting future research direction.
>
> ### 3. Regarding the choice of $\omega$
> For ensemble of $K=5$ reward models, **$\omega=1.0$ for the simplest toy case.**
> Based on our experiments, we speculate that **the appropriate scale is doubled (roughly) with 1) significant scale growth in terms of data size and 2) the introduction of new modality (e.g. unconditional or class-conditional $\to$ text-conditional model).**
> We use $\omega=2.0$ for Sections 5.2 and 5.4, where the data size grows to 256$\times$256 scale.
> For the additional Stable Diffusion experiment, we again double it and use $\omega=4.0$.
>
> Note that for the experiments of Section 5.3 (ImageNet Tench) or ablation studies with "Single" and "Union" models where we do not use ensemble, it is conceptually "fair" to use $K$ times larger value of $\omega$ in the following sense.
> If $r_\psi = \prod_{k=1}^K r_{\psi_k}^{(k)}$, we have $\nabla \log r_\psi = \sum_{k=1}^K \nabla \log r_{\psi_k}^{(k)}$.
> So viewing each $r_{\psi_k}$ as an approximation to $r$ (the "true" reward), $\nabla \log r_\psi \approx K \nabla \log r$ is used in the guidance process.
> Thus, to produce a similar effect, one should multiply $K$ when using log gradient from a single reward model.
> In all ablation studies, we stick to this rule.
> Similarly, in Section 5.3, we use $\omega=5.0$ which is $K$ times the value of $\omega$ used in Section 5.1.
>
> ### 4. Regarding censoring multiple concepts
> We thank the reviewer for mentioning this is interesting point.
> We believe that multiple concepts can be censored simultaneously by combining reward models corresponding to each one.
> Suppose we wish to censor two concepts, and for each of them we train a reward model $\hat{r}_1$ and $\hat{r}_2$ where $\hat{r}_i \approx 0$ indicates that the image contains (is malign with respect to) the $i$th concept.
> Similar to using the product in reward ensemble, we have $\hat{r}_1 \hat{r}_2 \approx 0$ if at least one of $\hat{r}_1, \hat{r}_2$ is small, so censored sampling guided by $\nabla \log \hat{r}_1 \hat{r}_2 = \nabla \log \hat{r}_1 + \nabla \log \hat{r}_2$ will enforce the generated images to be free of both concepts.

---

> > ### Comment · Reviewer_zmCo · 2023-08-12
> >
> > Thank you for the responses.

---

### Official Review · Reviewer_W62Q · 2023-07-12

**Soundness:** 3 good
**Presentation:** 3 good
**Contribution:** 3 good
**Rating:** 8
**Confidence:** 4

**Summary:**

The authors combine pre-trained diffusions with classifiers trained on human feedback about which type of images to omit, and use the classifiers to guide diffusion sampling using the Universal Guidance technique. They observe that they are able to filter out several types of malign images on a variety of datasets, notably removing undesirable artifacts on LSUN and faces on Imagenet.

I would recommend this paper for acceptance.

**Strengths:**

- People care about censoring sampled outputs, and don't like re-training large diffusions
- Together with the appendix, the specifics of how exactly to guide are well explored in this work
- I like that that while "bad" images are hard to quantify with some kind of numerical per-sample score, it is easy for a human to recognize instances of it when they do see it.
- emphasizing that not much feedback is needed for the cases explored

**Weaknesses:**

The main concerns would naturally be:
- how much feedback is enough
- are the good results specific to the type of things being censored in the particular datasets chosen
- for the infrequent malign content case, can a second dataset be used in place of model samples?

See questions below.

Other small comment: change "man hours" to "human work hours"

**Questions:**

For the first concern, it would be great if you could include something showing the number of model-produced malign samples going down as a function of number of malign samples used in classifier training, or as a function of number of total samples used for classifier training, or both. I know this is hard to do since the outputs need to be checked manually. Even doing this with a batch of 128 for a few classifier-training-set-sizes would be good, and see e.g. the malign proportion lowers for several feedback dataset sizes.

For the second concern, the crossed 7's are distinct enough in MNIST such that a very small classifier could quickly assign zero weight to them. However, the faces on Imagenet are more convincing since the properties to be censored (faces) do appear in several ways that aren't as clearly disentangled from benign properties. It would be useful to share somewhere in the text a case where the censoring did not work despite what looks like an adequate amount of samples. Did this happen for censoring some other of the Imagenet classes? Are there any cases you explore where the choice of malign samples constitute nearly all of the training set?

Feedback on a second dataset rather than model samples: for the case where a diffusion is trained on a dataset A with low frequency of the malign property (but nevertheless we need to guarantee that samples do not ever contain the property), the model samples will feature malign content infrequently. What if you have a second dataset B where the property is common? If the datasets are the same resolution and are diverse enough, and share some concepts this might work and might be a nice way to get past low malign frequency in dataset A? Consider the case where A only has some faces but B is at least half human faces. So samples from the A-trained-model do not produce malign content often and good feedback is hard to collect. How well does a feedback model trained on a second dataset B (which are not model samples) work for censoring the A-trained-diffusion from generating faces?


**Limitations:**

Addressed.

---

> ### Author Rebuttal · Authors · 2023-08-10
>
> We are delighted to see that the reviewer empathizes with our problem statement and solutions. We also greatly appreciate the inspiring comments with the positive evaluation. We have devoted our best efforts to provide satisfactory resposnes to each of the reviewer's concerns below.
>
> ### 0. Regarding the expression "man hours"
> We have replaced the instances of the expression "man"' to "human".
> We thank the reviewer for pointing this out.
>
> ### 1. Regarding the effect of feedback data size on malign proportion
> Reflecting the comment, we tested the effect of feedback data size used for reward training on the censoring performance in the setup of Section 5.3.
> Please refer to Figure 3b within the attached document in the common response.
> To clarify, we already had comparisons of the cases using 10, 20 and 30 malign images within the ablation studies, but newly added the cases of 50 and 100 malign images for clearer demonstration, and we do obtain a figure of the desired shape.
> We did not have enough capabilities to repeat the analogous procedure for the more complex setups of Section 5.4 and the additional Stable Diffusion experiment within the rebuttal period, but we plan to include them in the revised version of the paper.
> We thank the reviewer for the suggestion.
>
> ### 2. Regarding whether the success of proposed methods is general
> We would like to clarify that **the tasks we consider in the paper are not the ones particularly curated for success.**
> That is, we did not extensively explore setups other than what show in the paper; rather, we fixed the tasks from the beginning and developed the general strategies that could consistently deliver the results.
> Therefore, we believe that **our approach will be applicable over wide range of datasets and tasks.**
> It is true, though, that there seem to be a minimum number of samples required for positive results; e.g., using 50 malign and 50 benign samples for the LSUN Bedroom task was not satisfactory enough.
>
> We speculate that if the proportion of malign images from the baseline model is extreme (say, 99%) then it will be very difficult to successfully censor them out, but we have not tried pushing through experiments on such setups.
> Our intuition on the censored generation is: it works by encouraging a model's benign behavior while suppressing the malign side, which is possible only when the model already possesses sufficient capability of producing benign images.
> Therefore in designing experiments, we only considered the tasks where the baseline model generates a reasonable proportion of benign images, and for all of these cases we managed to achieve good censoring precision.
> In the later version of the paper, we will try to include more discussion on this point.
>
> ### 3. On using a secondary dataset
> We believe **using a secondary dataset is indeed a feasible strategy, and we provide a proof-of-concept experiment** using the LSUN Church setup where we censor the "Shutterstock" watermarks.
> Instead of marking samples generated by the model as malign, we visit Shutterstock website and manually collect 30 images (the number equal to the experiments from Section 5.2) within the search results for "church", "gothic church" and "cathedral" with clearly visible watermarks, using the Snipping Tool (please see Figure 2a of the attached document).
> We fix this secondary set of malign images $\mathcal{M}$ and train 5 reward models each using $\mathcal{M}$ together with 30 fresh random benign samples generated from the model.
> We use the same configurations as described in Section 5.2 and the Appendix I of the main paper for reward training.
> We observe that although slightly worse compared to the best ordinary censoring results from the paper **(0.76%)**, censoring via ensemble of the reward models trained using the secondary source is effective (achieving **1.4%** malign proportion) when used with $R=4$ recurrence steps (Table 2 in the attached pdf).
>
> On the other hand, it seems that this alternative approach should be used with more care.
> Our censored generation based on the secondary data does produce legitimate images with lowered proportion of malign images in the best case (pdf Figure 2b), but we heuristically observe frequent degradation in the generated images' quality without universal guidance (recurrence).
> Additionally, we observe that the guidance procedure becomes relatively less resilient to larger guidance weights when using the secondary data; using $\omega=2.0$ as in the paper seemed to lower the image quality, so we use $\omega=1.0$ for the results of the pdf's Table 2 and Figure 2b.
> This is possibly because it becomes much easier for the reward model to discriminate between the malign and benign images when malign images originate from a dataset of distinct distribution, so that it is more likely that the reward overfits and quickly learns to classify with extremely high confidence, causing the gradient to overshoot.
> Finally, if the secondary dataset is not sufficiently close to the distribution the diffusion model has learned, then censoring may be unsuccessful.
> When we try the same experiment using slightly different search keywords such as "old city", it does not work.
> This means that it is difficult to make a precise prediction on whether the strategy will succeed in, e.g., the virtual scenario suggested by the reviewer; it is genuinely case-by-case so the best practice seems to be trying out with hope.

---

### Official Review · Reviewer_ADeQ · 2023-07-21

**Soundness:** 3 good
**Presentation:** 2 fair
**Contribution:** 3 good
**Rating:** 6
**Confidence:** 2

**Summary:**

This paper proposes an approach to censor the sample generation of pre-trained diffusion probabilistic models to better align with human preferences. The authors use minimal human feedback (<3min spent in providing the feedback for basic tasks, and <15min for more complicated tasks that they consider) to train a light-weight reward model, then use the trained reward model to provide either time-dependent or time-independent guidance for the generation process. The authors also use techniques such as ensemble, iterative imitation learning, transfer learning, backward guidance and recurrence to improve censoring performance. Experiments on MNIST, LSU church, ImageNet, and LSUN bedroom showed the effectiveness of their proposed method.

**Strengths:**

The paper studies an interesting and important problem of aligning diffusion models with human preferences. By training a relatively lightweight reward model to guide the generation process instead of fine-tuning the large pre-trained model, their proposed approach could potentially save a lot of computation and human feedback.

**Weaknesses:**

- The paper's structure could be more coherent and consistent. The authors introduce classifier guidance in Section 1.1 but leave its relation to censored sampling unclear until Section 4. This disconnect could confuse readers. Similarly, there is a lack of clarity regarding the reward function $r$ and the reward model $r_\phi$ introduced in Section 2. While initially, it seems that $r_\phi \approx r$, the possibility of $r_\phi$ becoming time-dependent, as mentioned in Line 97 of Section 3, introduces confusion because $r$ is time-independent. This is not resolved until Section 4, when time-dependent guidance is addressed. To improve readability, it would be beneficial for the authors to introduce the underlying notations and mathematical concepts of the entire model—from the diffusion probabilistic models to the reward models and their role in censored sampling—before discussing training and evaluations.
- In section 3.1, the authors choose $r_\phi$ as the product of $K=5$ independently trained reward models. However, they did not have enough discussions or experiments to back up these choices, which makes the resulting model seem arbitrary. In particular, since the authors mentioned $r_\phi(X)\approx r(X)=P(Y=1|X)$, it's unclear why taking the product--as opposed to taking the average--makes sense because each $r_{\psi_k}$ is also approximating $r$. Moreover, if we ignore the mathematical implications for now, the authors explained in line 115 that taking the product is essentially asking for unanimous approval, but readers might still question whether other methods, such as increasing the value of $w$ in the "union" method, could work just as well.
- The paper highlights extreme human feedback efficiency by saying that a few minutes of human feedback are sufficient. However, this claim overlooks the complexity of tasks considered in this paper, which are relatively simple compared to tasks where RLHF is typically employed in large-scale models. Therefore, it would be better to directly compare with previous approaches (such as fine-tuning) under the same task. Additionally, the more complex tasks, like censoring distorted bedrooms, actually require more than the estimated 3 minutes of human feedback. Hence, the title could be seen as potentially misleading or exaggerating the method's efficiency.

**Questions:**

- For the training data $X^{(1)},\cdots,X^{(N)}$ in Algorithms 1 and 2, are they images corrupted by the VP SDE or images generated by the pre-trained model $\varepsilon_\theta$? What are the objectives in training each reward model in the ensemble method?
- In Algorithm 1, for each $k$ the learner randomly select with replacement $N_M$ benign samples. Does this step also need human feedback?
- Typo in line 204. "30 malign and 150 malign samples"

**Limitations:**

Yes

---

> ### Author Rebuttal · Authors · 2023-08-09
>
> We thank the reviewer for constructive comments. We made our best efforts to address the concerns and reflect the comments to improve the paper. For the concerns regarding the complexity of the tasks we cover, comparison against other methods, and the longer human time for the LSUN Bedroom task, please refer to the pertaining discussions within our common response. We hope that we have properly addressed the reviewer's concerns, and if that is the case, we kindly ask the reviewer to consider increasing the rating.
>
> ### 1. On the presentation of the paper
> We highly appreciate these suggestions. We will clarify in Section 1.1 that classifier guidance is the basis of our censoring framework and is therefore relevant to the paper's contents. We will also clarify within Section~2 that the reward function $r(\cdot, t)$ can be time-dependent and considering the time-dependent reward helps, in some setups, to properly deal with images with different level of noises that are encountered during the sampling process of diffusion models.
>
> ### 2. On the choice of reward model
> Indeed, there are multiple ways to combine individual reward models $r_{\psi_k}^{(k)}$ to build a refined approximation of the true reward $r$, and in our initial experiments (not included in the paper), we tested each of
> - Reward averaging: using $r_\psi = \frac{1}{K} \sum_{k=1}^K r_{\psi_k}^{(k)}$
> - Logit averaging: using $r_\psi = \sigma \left( \frac{1}{K} \sum_{k=1}^K h_{\psi_k}^{(k)} \right)$, where $r_{\psi_k}^{(k)}(X) = \sigma \left( h_{\psi_k}^{(k)}(X) \right)$ for each $k=1,\dots,K$ and $\sigma$ is the sigmoid function
> - Geometric reward averaging: using $r_\psi = \left( \prod_{k=1}^K r_{\psi_k}^{(k)} \right)^{1/K}$.
>
> While each method seems to make sense in principle, it turned out that methods 1, 2 were not as empirically effective as method 3 in censoring out the unwanted samples.
> Note that taking the product of each $r_{\psi_k}^{(k)}$ is equivalent to using method 3 and then increasing the guidance weight $\omega$ by $K$ times.
> We would like to clarify that **we already did make a fair comparison, in this context, between the "ensemble" models versus the "union" models** in all of our ablation studies in Sections H.3, I.4 and K.4, where we used $K$ times larger values of $\omega$ for the "union" cases compared to the "ensemble" cases, which are the product models. Our experiments conclude that ensemble models are still always better.
>
> One must use the guidance weight $\omega = 1.0$ to strictly obey the mathematical formalism $p_{\mathrm{censor}}(x) = p_{\mathrm{data}}(x) r(x)$, but we believe it is an everyday practice in the diffusion model literature to use larger $\omega$ for improved consistency with the conditioning label in guided/conditional generation.
> In the end, we observed that using the product of $K=5$ reward models, which is equivalent to using $\omega = 5.0$ with respect to the geometric average, already works fairly well in most cases even if it is not a carefully tuned choice. (We do expect further performance gain by ensembling a larger number of reward models, but we do not pursue this minor improvement direction in depth.)
> For the interest of space, we omitted these details and only mentioned that we use the product as our reward.
> However, we would be happy to further justify this choice by including the above discussion, if we are later provided with additional space.
>
> ### 3. Question regarding the notation $X^{(1)}, \dots, X^{(N)}$ and reward model training
> $X^{(1)}, \dots, X^{(N)}$ are the images generated by the pre-trained model, and in Algorithm 1, we assume that the human feedback (labels for malign/benign) is already given on them.
> When using the time-dependent reward model architecture, we also sample $t$ uniformly within $[0, T]$ and noise $X^{(i)}$ up to time $t$ along the VP SDE.
> We use the weighted BCE loss function as described in our Appendix, Section F: $BCE_\alpha(r_\psi(x;t), y) = -\alpha \cdot y \log r_\psi(x;t) - (1-y) \log (1-r_\psi(x;t))$ where $\alpha < 1$ is a hyperparameter, determining to which extent the reward model prioritizes classifying malign images as malign over benign images as benign.
>
> ### 4. Question on the subsampling of benign samples
> In the setup of Algorithm 1, we assume that benign images are more frequently generated compared to malign images.
> The human feedback collecting-stage terminates once the number of malign samples reaches the desired level ($N_M$), and at this moment one will already have a pool of abundant ($N_B > N_M$) benign samples (because the feedback provider is automatically marking images as benign if they are not malign).
> Thus, during the execution of Algorithm 1, no additional human feedback is required.

---

> > ### Comment · Reviewer_ADeQ · 2023-08-15
> >
> > I appreciate the author's rebuttal and the additional experiment on stable diffusion. Given that most of my concerns have been addressed, I will raise my score from 4 to 6.

---

### Author Rebuttal · Authors · 2023-08-09

# Common Response (pdf attached)

We thank all reviewers for the extremely detailed and constructive feedbacks. We are delighted that most reviewers found our ideas convincing and the contributions solid. Below, we provide our response to some common concerns.
### 1. List of additional experiments
In the attached, we show 4 additional experiments.
For better readability, we provide the list of the experiments associated to each figure/table.
- Figure 1: A new task using **Stable Diffusion**. We successfully eliminate embedded raw texts that are unexpectedly associated to the prompt "a photo of a human" (Table 1). This experiment is to address the concerns (reviewers ADeQ, 6Ndg) that our framework may not apply/not be as efficient as we claim when diffusion models are of larger scale & complexity. Please refer to Section 2 below for details.
- Table 2: Censoring result using **malign images from a secondary source** (instead of model-generated samples) in the LSUN Church task. This demonstrates that external images may substitute model-generated malign images if they are scarce. We thank reviewer W62Q for motivating this experiment.
- Figure 3a: **Round 4 of imitation learning** in the ImageNet task provides further improvement over Round 3. The continued application of imitation learning reduces the malign proportion approximately exponentially ($y$-axis is in log scale) and could lead to extreme precision as reviewers inquired (GFro, iXGj).
- Figure 3b: Plot highlighting the effect of feedback size for reward training (ImageNet task), as suggested by reviewer W62Q.

### 2. Regarding application to complex tasks/large-scale models
To resolve the incorrect impression that our methods are efficient only for simple settings, we introduce a **Stable Diffusion experiment that uses minimal human efforts.**

In Stable Diffusion v1.4, **21.9%** out of 1,000 generated images (512$\times$512) by the prompt "a photo of a human" contain prominent embedded texts, or even lacks any genuine image and displays only the raw text (Figure 1a).
(We could not find a straightforward modification to the prompt that removes texts without compromising the generality of the prompt; basic attempts like adding "without text" or "no text" are unsuccessful.)
We collect human feedback data until 100 malign images are found **(3~4 mins)**.
We train 5 reward models via transfer learning (batch size 64, no augmentation, $BCE_\alpha$ weight $\alpha=0.1$, lr $10^{-4}$ and 10,000 iterations).
We use $\omega=4.0$ and no backward guidance.
Without/with recurrence ($R=4$), the proportion of malign samples drops to **4.8%**/**1.3%** (out of 1,000 samples).
Figure 1b shows generation results with all censoring techniques applied.
We hope this resolves some reviewers' concern regarding applicability on large-scale text-to-image models.

### 3. Regarding the human time for LSUN bedroom task
We do acknowledge natural criticisms from some reviewers that this task actually requires $>$3 minutes. However, we point out that
- This is a large-scale task: it uses the full 256$\times$256 diffusion model (not the lighter LDM).
- In fact, we design this setup as a **worst case** on the human work our framework requires.
Most censoring tasks in practice are likely to be less delicate than this task, where we deal with an ambiguous concept of malignity that is difficult to characterize and inevitably requires multiple guidelines to minimize subjectivity of the experiments.
- In the end, even the amount of feedback ($<$1k) and human time (15 mins) for this worst case is significantly smaller compared to those in prior works [1, 2, 3] that use 27~137k human feedback to fine-tune diffusion models.
- Nevertheless, we'll update the title to "Censored Sampling ... Using **a Few Minutes** of Human Feedback".

### 4. Comparison to other censoring methods
To clarify, our paper aims to resolve a defect/inconsistency of a model whose occurrence is **easy for humans to identify, but difficult to be mathematically described or fixed**.
There are many issues of this kind, e.g.: NSFW contents, inconsistencies in limbs/fingers, or double-headed humans, persistently appearing even after negative prompting.
Our premise is that **there are cases in which feedback from humans are the only simple or feasible way of encoding the model's defect, and for these types of tasks, any other methodologies such as fine-tuning should share the process of collecting human feedback data and learning from these data**.

Traning a reward model fitting the human feedback data and then fine-tuning as in [1, 2, 3] involves more complications compared to our approach, which is arguably one of the simplest things to do with a reward model.
Fine-tuning usually requires multiple trial-and-errors until the best hyperparameters are found, as the diffusion model often overfits to the fine-tuning objective and experiences catastrophic forgetting.
(This entire process is not necessary if we simply use guidance.)
Another possible approach is to apply the policy gradient type fine-tuning as in [4] without explicitly training the reward model but letting humans play the role of the environment (reward oracle).
However, we see less hope in this direction because policy gradient is notoriously sample inefficient [5], unless technical endeavor comparable to the one devoted to this whole paper is made.
**Therefore, we argue that other approaches, including those based on fine-tuning, may not be as straightforward or suitably sample-efficient as they might seem at first glance.**

[1] Lee et al., Aligning Text-to-Image Models using Human Feedback. 2023.

[2] Xu et al., ImageReward: Learning and Evaluating Human Preferences ... Generation. 2023.

[3] Fan et al., DPOK: Reinforcement Learning for Fine-tuning ... Diffusion Models. 2023.

[4] Fan & Lee, Optimizing DDPM Sampling with Shortcut Fine-Tuning. ICML, 2023.

[5] Gu et al., Q-Prop: Sample-Efficient Policy Gradient ... Critic. ICLR, 2017.

---

> ### Author Response · Authors · 2023-08-21
>
> We sincerely thank all reviewers for their time and effort spent for reviewing our paper.
> We believe the valuable comments and discussions from the rebuttal period have significantly improved the paper, and we will adequately incorporate them into the revised version.

---

### Decision · Program_Chairs · 2023-09-21

**Decision:**

Accept (poster)

**Comment:**

All the reviewers agree with the novelty and contribution of the paper. Thus I recommend acceptance.